# Motor unit dysregulation following 15 days of unilateral lower limb immobilisation

Thomas B. Inns[1], Joseph J. Bass[1], Edward J.O. Hardy[1,2], Daniel J. Wilkinson[1], Daniel W. Stashuk[3], Philip J. Atherton[1], Bethan E. Phillips[1] and Mathew Piasecki[1]

[1]*Centre Of Metabolism, Ageing & Physiology, MRC-Versus Arthritis Centre for Musculoskeletal Ageing Research and NIHR Nottingham BRC, University of Nottingham, Derby, UK*
[2]*Department of Surgery and Anaesthetics, Royal Derby Hospital, Derby, UK*
[3]*Department of Systems Design Engineering, University of Waterloo, Ontario, Canada*

Handling Editors: Scott Powers & Kevin Murach

Linked articles: This article is highlighted in a Perspective article by Soendenbroe. To read this article, visit https://doi.org/10.1113/JP283800.

The peer review history is available in the Supporting Information section of this article (https://doi.org/10.1113/JP283425#support-information-section).

**Abstract** Disuse atrophy, caused by situations of unloading such as limb immobilisation, causes a rapid yet diverging reduction in skeletal muscle function when compared to muscle mass. While

**Thomas Inns** is a PhD student at the Centre of Metabolism, Ageing and Physiology at the University of Nottingham, investigating the impact of disuse atrophy on facets of muscle function. Primarily focused on neuromuscular physiology, this interest has extended into ageing and exercise across different muscle groups and cohorts to contribute to this rapidly developing field.

B. E. Phillips and M. Piasecki share co-senior authorship.
This article was first published as a preprint. Inns TB, Bass JJ, Hardy EJO, Wilkinson DJ, Stashuk DW, Atherton PJ, Phillips BE, Piasecki M. 2022. Motor unit dysregulation following 15 days of unilateral lower limb immobilisation. bioRxiv. https://doi.org/10.1101/2022.06.01.494421.

mechanistic insight into the loss of mass is well studied, deterioration of muscle function with a focus towards the neural input to muscle remains underexplored. This study aimed to determine the role of motor unit adaptation in disuse-induced neuromuscular deficits. Ten young, healthy male volunteers underwent 15 days of unilateral lower limb immobilisation with intramuscular electromyography (iEMG) bilaterally recorded from the vastus lateralis (VL) during knee extensor contractions normalised to maximal voluntary contraction (MVC), pre and post disuse. Muscle cross-sectional area was determined by ultrasound. Individual MUs were sampled and analysed for changes in motor unit (MU) discharge and MU potential (MUP) characteristics. VL CSA was reduced by approximately 15% which was exceeded by a two-fold decrease of 31% in muscle strength in the immobilised limb, with no change in either parameter in the non-immobilised limb. Parameters of MUP size were reduced by 11% to 24% with immobilisation, while neuromuscular junction (NMJ) transmission instability remained unchanged, and MU firing rate decreased by 8% to 11% at several contraction levels. All adaptations were observed in the immobilised limb only. These findings highlight impaired neural input following immobilisation reflected by suppressed MU firing rate which may underpin the disproportionate reductions of strength relative to muscle size.

(Received 14 June 2022; accepted after revision 19 August 2022; first published online 11 September 2022)

**Corresponding author** Mathew Piasecki: Centre Of Metabolism, Ageing & Physiology, MRC-Versus Arthritis Centre for Musculoskeletal Ageing Research and NIHR Nottingham BRC, University of Nottingham, Derby, UK.    Email: mathew.piasecki@nottingham.ac.uk

**Abstract figure legend** Ten healthy young males underwent 15 days of unilateral lower limb immobilisation with an irremovable leg brace. Muscle size, strength, and neuromuscular characteristics were measured bilaterally. Muscle strength reduced to a greater extent than muscle size in the immobilised leg while remaining unaltered in the non-immobilised leg. Motor unit firing rate, measured bilaterally using intramuscular electromyography, was also reduced in the immobilised leg only. This occurred at contraction intensities both relative to follow-up muscle strength and muscle strength normalised to pre-immobilisation. These findings suggest that neural dysregulation contributes to the loss of muscle strength observed in situations of disuse atrophy in humans.

## Key points

- Muscle mass and function decline rapidly in situations of disuse such as bed rest and limb immobilisation.
- The reduction in muscle function commonly exceeds that of muscle mass, which may be associated with the dysregulation of neural input to muscle.
- We have used intramuscular electromyography to sample individual motor unit and near fibre potentials from the vastus lateralis following 15 days of unilateral limb immobilisation. Following disuse, the disproportionate loss of muscle strength when compared to size coincided with suppressed motor unit firing rate.
- These motor unit adaptations were observed at multiple contraction levels and in the immobilised limb only. Our findings demonstrate neural dysregulation as a key component of functional loss following muscle disuse in humans.

## Introduction

Disuse atrophy is the loss of skeletal muscle mass associated with decreased external loading or complete immobilisation. It commonly occurs following joint trauma, nerve injury, or prescribed bed rest (Bodine, 2013), progressing rapidly with reductions of strength occurring after just 5 days of immobilisation (Wall et al., 2014). As such, it has been widely applied as an experimental model to investigate underpinning mechanisms in scenarios of space flight, prolonged bed rest, spinal cord injury, and ageing (Castro et al., 1999; Narici & De Boer, 2011; Puthucheary et al., 2013). The trajectory of decline is not linear; five days of unilateral lower limb suspension lead to ∼ 3% reduction in quadricep cross-sectional area (CSA) (Wall et al., 2014) and eight weeks of bed rest reduced quadriceps CSA by ∼ 14% (Mulder et al., 2006).

The loss of muscle strength with disuse is commonly reported to exceed the loss of muscle size. Following 8 weeks of bedrest, quadriceps CSA declined by 14% compared to a 17% decline in strength (Mulder et al., 2006), while only 10 days of bed rest was enough to elicit a $\sim$ 6% reduction in CSA and a $\sim$ 14% reduction in knee extensor maximal voluntary contraction (MVC) (Monti et al., 2021). Furthermore, meta-analysis of bed rest studies with a combined 118 participants across various durations of disuse found no relationship between length of bed rest and reductions in muscle size, whereas muscle power reduction and bed rest duration were strongly related (Di Girolamo et al., 2021). Numerous data provide mechanistic insight to muscle *atrophy*, including decreased MPS, increased MPB, mitochondrial dysfunction, insulin resistance, and histochemical markers of fibre denervation (Monti et al., 2021; Phillips et al., 2009; Rudrappa et al., 2016), yet mechanistic insight into loss of muscle *function* is less clear.

Successful activation of muscle relies upon coordinated input to the motoneuron pool and synaptic transmission across neuromuscular junctions (NMJs), with increases in muscle force mediated by the recruitment and rate modulation of functioning motor units (MUs). Motor units from hand muscles showed a marked reduction in firing rate (FR) following disuse, which corresponded with reduced force generating capacity (Duchateau & Hainaut, 1990). Findings from the vastus lateralis (VL) following 10 days of bed rest showed an increased number of NCAM positive fibres, suggesting a contribution of fibre denervation following disuse (Monti et al., 2021). Furthermore, surface electromyography (EMG) amplitude following 14 days of unilateral lower limb suspension (ULLS) decreased in the immobilised limb along with peak torque, while limb mass remained unchanged (Deschenes et al., 2002), all of which highlight potential neural dysregulation. The extent of disuse atrophy is also muscle dependent which has led to the descriptors of 'atrophy resistant' and 'atrophy susceptible' muscles (Bass et al., 2021), with the quadriceps deteriorating at a greater rate than the hamstrings following 7 days of immobilisation (Kilroe et al., 2020). A recent systematic review consolidated previous work to date (Campbell et al., 2019), and of the 40 studies included, 20 immobilised the knee, preventing use of a functionally relevant muscle group, i.e. the quadriceps. Furthermore, just 20 studies investigated neural factors (i.e. using surface EMG), yet adaptations of individual MU characteristics, particularly at the NMJ, remain largely unexplored.

Intramuscular EMG (iEMG) is a minimally invasive technique which enables the *in-vivo* recording of individual MU potentials (MUP) during voluntary muscle activation, and as such has the potential to reveal functional and structural adaptations of MUs at a range of contraction levels (Piasecki, Garnés-Camarena et al., 2021). The purpose of this study was to quantify within-subject VL neuromuscular adaptation of immobilised and non-immobilised limbs pre and post 15-days of unilateral immobilisation. We hypothesised that, coinciding with a reduction in muscle mass and strength, electrophysiological markers of NMJ transmission instability would increase, and MU firing rate would decrease in the VL of the immobilised limb, with no change to the non-immobilised limb.

## Methods

### Ethical approval

This study was approved by the University of Nottingham Faculty of Medicine and Health Sciences Research Ethics Committee (103-1809) and conformed with the Declaration of Helsinki. It was registered online at clinicaltrials.gov (NCT04199923). Participants aged 18 to 40 were recruited locally from the community via advertisement posters in print and on research group social media pages. Ten healthy, young male participants were recruited to take part in the study. After providing written informed consent to participate in the study, potential participants were screened for eligibility against pre-determined exclusion criteria, including body mass index (BMI) >18 or <35 kg·m$^2$, active cardiovascular, cerebrovascular, respiratory, renal, or metabolic disease, active malignancy, musculoskeletal or neurological disorders. All participants were recreationally active, and one was an active smoker, and no participants were regularly using nutritional supplements. Once eligibility was confirmed, participants were invited to the laboratory for baseline testing, as described below (Fig. 1*A*).

### Muscle ultrasound

VL cross sectional area scans (CSA) were taken at the mid-belly of the muscle (Fig. 1*D*). This was located by measuring the length from the midline of the patella to the greater trochanter and taking the middle value of that line. Following this, a narrow ultrasound probe (LA523 probe and MyLab$^{TM}$50 scanner, Esaote, Genoa, Italy) was used to find the medial and proximal borders of the muscle where the aponeurosis of the VL intersected with the *vastus intermedius*. Three axial plane images were collected following this line from both legs and subsequently analysed using ImageJ (Laboratory of Optical and Communication, University of Wisconsin-Madison, WI, USA) to quantify CSA (Scott et al., 2017). Three measurements were made from each image which were

averaged, providing three mean values per participant that were subsequently averaged for each leg and timepoint for analysis. The same operator performed and analysed all scans to reduce inter operator bias. Due to equipment malfunction at follow-up, $n = 9$ for US.

### Lower limb power

Explosive unilateral lower limb power was assessed using the Nottingham Power Rig (University of Nottingham; (Bassey & Short, 1990)). For this, participants were seated on the purpose-built rig with their knee at 90° when

the foot was on the plate. The rig is designed to isolate power production from the lower limb. Following a 3 s countdown, participants were instructed to perform an explosive push. The power exerted was displayed digitally and the highest of three attempts recorded. Due to equipment malfunction at follow-up, $n = 9$ for unilateral lower limb power assessment.

### Maximal voluntary isometric contraction

Participants were seated in a custom-built isometric dynamometer (Load cell amplifier; LCA1, 12V 1A medical

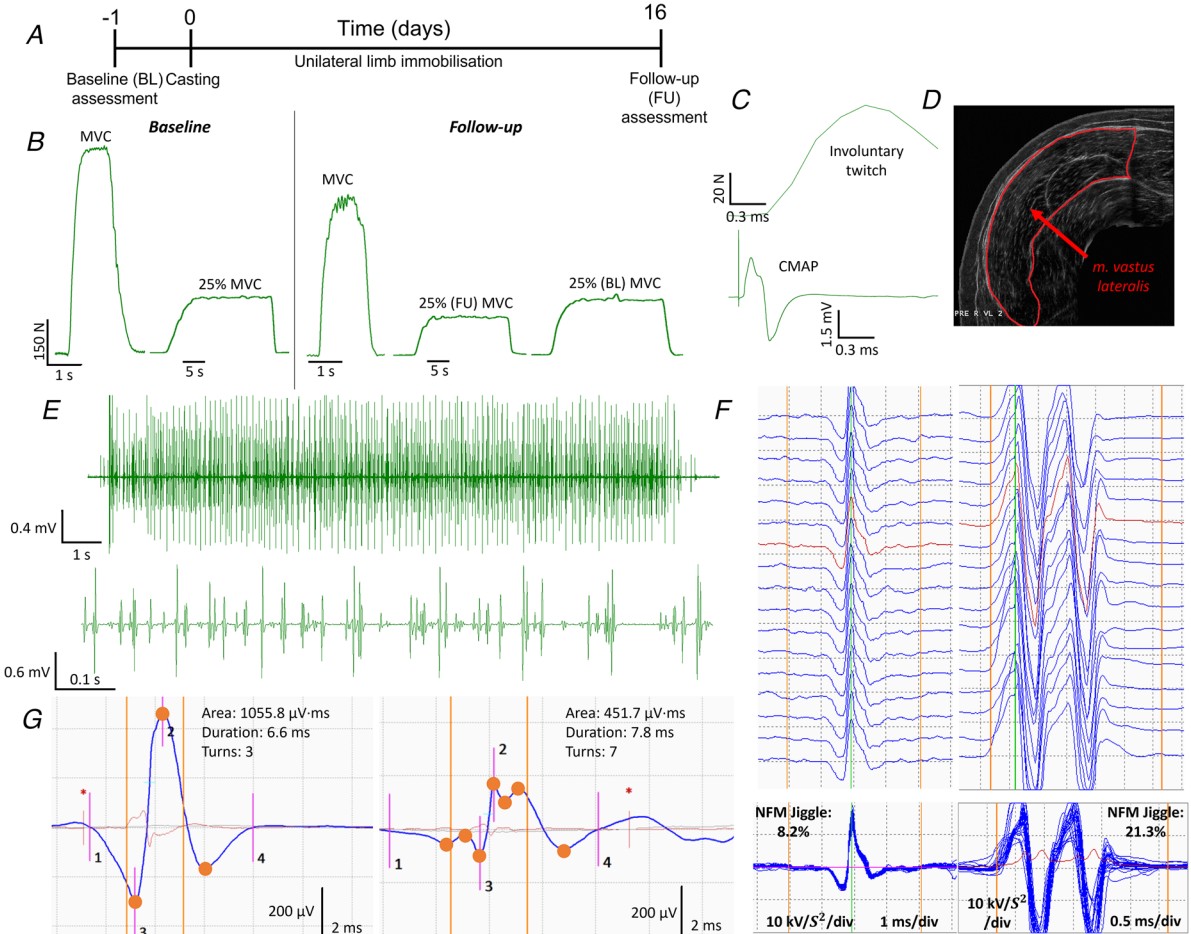

**Figure 1. Methodological outline**
*A*, schematic of the study timeframe. *B*, knee extensor force traces (left to right) from a maximal voluntary isometric contraction (MVC), 25% of MVC contraction pre 15-day unilateral limb immobilisation, 25% contraction post-immobilisation relative to follow-up MVC, and 25% contraction post-immobilisation relative to baseline MVC. *C*, peak twitch force recording showing force trace (upper) and corresponding surface electromyography EMG recorded M-wave (lower) from a maximal stimulation of the femoral nerve. *D*, an example ultrasound scan of the m. vastus lateralis used to calculate cross-sectional area. *E*, example intramuscular EMG (iEMG) trace recorded during a 12 s voluntary isometric contraction (upper) and magnified (lower) with visible motor unit potentials (MUP). *F*, example raster plots (upper) and corresponding shimmer plots (lower) from near-fibre MUPs (NF-MUP) before (left) and after (right) immobilisation recorded during active contractions. *G*, example MUPs recorded before (left) and after (right) immobilisation. Turns are marked with an orange circle. N; newtons, s; seconds, ms; milliseconds, V; volts, $\mu$V; microvolts, kV/s$^2$; kilovolts per second squared, $\mu$V·ms; microvolts per millisecond.

PSU, GDM25B12-P1J, Sunpower Electronics, Reading, UK) with their knee joint angle fixed at 90° while the hip joint angle was at 110°. The ankle was secured in place to a plate connected to a force transducer. Three moderate intensity warm-up contractions were carried out with visual feedback of force traces on a screen in front of the participants. Using a waist belt to prevent hip lifting and facilitate isolation of the knee extensors, participants were verbally encouraged to perform an isometric knee extension at maximal capacity. Three attempts were carried out and the highest value was taken as the maximal and used to determine voluntary contraction intensity (Fig. 1*B*).

### Vastus lateralis motor point identification

Using low intensity percutaneous electrical stimulation (400V, pulse width 50 $\mu$S, current ~10 mA; delivered via a Digitimer DS7A, Welwyn Garden City, UK), the surface of the VL was explored to find the point producing the greatest visible twitch with the lowest current, i.e. the motor point (Piasecki, Ireland, Jones et al., 2016).

### Force and electromyography recording

Surface electromyography was arranged in a bipolar configuration. The recording electrode was placed over the identified motor point and the reference electrode over the patellar tendon (disposable self-adhering Ag-AgCl electrodes; 95 mm$^2$; Ambu Neuroline, Baltorpbakken, Ballerup, Denmark). A ground electrode was placed just above the reference electrode on the patella (Ambu Neuroline Ground). EMG signals were digitised (CED Micro 1401; Cambridge Electronic Design, Cambridge, UK) and Spike2 (version 9.00a, CED) software was used to provide a real-time display of the signal on screen. Peak twitch force was calculated using percutaneous electrical stimulation (Digitimer, UK), and the maximal M-wave was measured with surface electromyography at the motor point. A stimulating pen was placed over the femoral nerve in the inguinal fold and stimulation intensity (400V, pulse width 50 $\mu$S, current typically 70 to 110 mA) was increased until the M-wave amplitude displayed in Spike2 plateaued. The peak force generated corresponding to the maximal M-wave was recorded, and all force signals were recorded at 100 Hz (Fig. 1*C*). As three participants were unable to tolerate femoral nerve stimulation at follow-up visits, $n = 7$ for this assessment. Force steadiness (FS) was assessed during the sustained voluntary contractions (described below) as the coefficient of variation of the force, averaged at each contraction intensity. The first two passes of the target (<1 s) were excluded from the calculation to avoid corrective actions when reaching the target line.

### Intramuscular electromyography

A concentric needle electrode (Ambu Neuroline model 740 25−45/25, Ambu, UK) was inserted at the *vastus lateralis* motor point. The recorded iEMG signals were sampled at 50 kHz and bandpass filtered from 10 to 10 kHz (1902 amplifier, CED, Fig. 1*E*). A real-time display was observed using Spike2 (CED) and data were stored offline for analysis. Once the needle was inserted, contractions were carried out at 10% and 25% of the participants MVC following a visual target line (Fig. 1*B*). Six contractions were recorded at each intensity, with the needle electrode position altered between each to sample a broader range of MUPs (Jones et al., 2021). From the original position, the needle electrode was slightly withdrawn, and the bevel rotated 180° at each depth and positioned to maximise signal to noise ratio. These voluntary contractions were held for 12 s with a ~20 s rest in between each contraction.

### EMG analysis

Decomposition-based quantitative electromyography (DQEMG) was used for all iEMG data analysis. This involved the detection of MUPs and extraction of MUP trains (MUPTs) from individual MUs generated during sustained iEMG signals recorded during voluntary contractions at set intensities. MUPTs were excluded if they contained MUPs from multiple MUs or fewer than 40 MUPs. MUP templates were visually inspected to ensure markers were placed correctly to start, end, positive, and negative peaks of the waveforms. MUP area was defined as the integral of the absolute value of MUP values recorded between start and end markers multiplied by the sampling time interval in $\mu$V/ms (Piasecki, Garnés-Camarena et al., 2021). MUP amplitude was determined as the measurement from the maximal positive and negative peaks of the waveform (Guo et al., 2022). MUP complexity, measured as the number of turns, was defined as the number of significant slope direction changes within the duration of the MUP of a height >20 $\mu$V (Fig. 1*G*). A near fibre MUP (NF-MUP) was obtained from each MUP by estimating the slopes of each MUP (Piasecki, Garnés-Camarena et al., 2021) and NMJ transmission instability, measured as near-NF-MUP jiggle, was determined as the normalised means of median consecutive amplitude differences (Piasecki, Garnés-Camarena et al., 2021) (Fig. 1*F*). MU FR was recorded as the rate of occurrence per second of MUPs within a MUPT in Hz, and MU FR variability was determined as the coefficient of variation of the inter-discharge interval.

### Immobilisation procedure

For the 15 days of immobilisation, a unilateral lower-limb immobilisation model was used. Although limb

immobilisation has been reported to impact muscle tone and cellular metabolism in rats (Booth, 1977; Goldspink et al., 1986; Hackney & Ploutz-Snyder, 2012), it was used in favour of ULLS in this study to prevent accidental weight bearing. The knee joint was fixed at 75° flexion using a hinged leg brace (Knee Post op Cool, Össur, Iceland) with the ankle joint fixed using an air-boot (Rebound Air Walker, Össur), ensuring that the immobilised leg was not able to bear any weight. Crutches were provided and adjusted according to the height of the participant, and training on their effective use was provided. The brace and boot remained in place at all times, including sleeping and bathing, with tamper tags attached to each to monitor intervention adherence.

### Follow-up testing

Following 15 days of immobilisation, participants were invited back to the laboratory for post-immobilisation testing (Fig. 1*A*). All procedures carried out in the baseline testing visit were repeated in both legs with iEMG performed at the same motor point as the initial visit. In addition to measuring FS and iEMG at 25% of post-immobilisation MVC (i.e. relative force, referred to as follow-up force), iEMG was also performed at 25% of pre-immobilisation MVC (i.e. absolute force, referred to as baseline force) to compare FS and MU characteristics normalised to pre and post disuse-induced strength loss.

### Statistical analysis

Statistical analysis of CSA, MVC, twitch force, power, and FS was performed using GraphPad Prism version 9.1.0 (GraphPad Software, CA, USA) using repeated measures 2-way analysis of variance with Šidak's post-hoc analysis in the event of a significant interaction. Multi-level mixed effect linear regression models were used to analyse MU parameters, in StataSE (v16.0, StataCorp LLC, TX, USA). For these models the first level was single motor unit; single motor units were clustered according to each participant to form the second level, which was defined as the participant level and reflects the total n. Two within-subject factors were included; leg (immobilised and control) and time (pre and post), and leg x time interactions were included in all models. Additional exploratory analyses were performed to investigate relationships between iEMG variables and key physiological outcome variables – muscle strength (MVC) and size (CSA) using R (Version 4.2.0, (https://cran.r-project.org/) implemented using R studio. Firstly, correlative analysis was assessed with Pearson's product moment correlation coefficient and visualised using corrplot (https://cran.r-project.org/web/packages/corrplot) to determine any

strong relationships between variables. Cluster analysis and principal component analysis for variables were performed using the ClustOfVar (https://cran.r-project.org/web/packages/ClustOfVar) and factoextra (https://cran.r-project.org/web/packages/factoextra) packages respectively, to determine which variables strongly clustered and related to others. The variables used for these analyses included those measured from 25% pre immobilisation and 25% follow-up in the immobilised limb only, namely MVC, CSA, MUP area, MUP amplitude, MUP complexity, NMJ transmission instability, MU FR and MU FR variability. Finally, using a subset of these best clustering variables, namely MVC, CSA, FR, FR variability, and NMJ transmission instability, multivariate linear regression was performed to determine which clustered variables best predict changes in MVC and CSA. Significance was assumed if $P < 0.05$.

## Results

### Participant characteristics

Ten male participants took part in this study. Characteristics are shown in Table 1.

### Muscle size and function

There was a significant leg x time interaction in VL CSA ($P < 0.001$), which decreased in the immobilised leg after 15 days ($-15\%$, $P < 0.001$) and remained unchanged in the control leg ($P = 0.761$, Fig. 2*A*). Similarly, there was a significant leg x time interaction of knee extensor MVC ($P = 0.005$), which decreased in the immobilised leg ($-31\%$, $P < 0.001$) while no change was seen in the control leg ($P = 0.498$, Fig. 2*B*). Peak twitch force showed no significant leg x time interaction ($P = 0.549$, Fig. 2*C*). Unilateral lower limb power output presented a significant leg x time interaction ($P = 0.017$), which reduced significantly in the immobilised limb ($-26\%$, $P = 0.003$) while remaining unchanged in the control leg ($P = 0.939$, Fig. 2*D*). Force steadiness (FS) at 10% MVC presented a significant leg x time interaction ($P = 0.020$, Fig. 2*E*), with non-significant changes in the immobilised

**Table 1. Descriptive characteristics of participants showing mean and standard deviation (SD)**

| $N = 10$ | Mean (SD) |
|---|---|
| *Age (years)* | 23.7 (3.4) |
| *Height (cm)* | 181.6 (6.7) |
| *Weight (kg)* | 79.4 (9.7) |
| *BMI (kg/m$^2$)* | 24.0 (2.1) |

limb (− 14%, $P = 0.077$) and the control ($P = 0.278$). No interaction was present in FS at 25% MVC ($P = 0.255$, Fig. 2*E*), nor was an interaction present in FS at 25% baseline MVC ($P = 0.349$).

### Neuromuscular characteristics

MUP area and amplitude at 10% MVC presented no significant leg x time interaction ($P = 0.091$ and $P = 0.106$, respectively, Figs 3*A* and *B*, 6*A* and *B*). MUP complexity, defined as the number of turns, presented no significant leg x time interaction at 10% MVC ($P = 0.660$, Figs 3*C* and 6*C*). At 10% MVC, NMJ transmission instability showed no leg x time interaction ($P = 0.191$, Figs 3*D* and 6*D*) Motor unit FR presented a significant leg x time interaction at 10% MVC ($P = 0.022$, Fig. 3*E* and 6*E*), with a reduction in the immobilised leg ($P < 0.001$) but no change in the control ($P = 0.848$). Firing rate variability presented no leg x time interaction at 10% ($P = 0.070$, Figs 3*F* and 6*F*).

MUP area at 25% follow-up MVC presented a significant leg x time interaction ($P = 0.028$, Fig. 4*A*

and 6*A*), reflected by a reduction in the immobilised leg ($P < 0.001$) but no change in the control ($P = 0.717$). Similarly, MUP amplitude presented a significant leg x time interaction ($P = 0.046$, Fig. 4*B* and 6*B*), with a reduction in the immobilised leg ($P = 0.007$) but no change in the control ($P = 0.881$). MUP complexity showed a significant leg x time interaction ($P = 0.007$, Fig. 4*C* and 6*C*), with an increase in the immobilised leg ($P = 0.009$), but no change in the control ($P = 0.600$). For NMJ transmission instability, as assessed by NF-MUP Jiggle, at 25% follow-up MVC, no leg x time interaction was observed ($P = 0.966$, Fig. 4*D* and 6*D*). For FR at 25% follow-up MVC a significant leg x time interaction was also observed ($P < 0.001$, Fig. 4*E* and 6*E*), again showing a reduction in the immobilised leg ($P < 0.001$) but no change in the control ($P = 0.563$). Firing rate variability at 25% follow-up MVC presented no significant leg x time interaction ($P = 0.082$, Fig. 4*F* and 6*F*).

At 25% baseline MVC, a significant leg x time interaction was seen in MUP area ($P < 0.001$, Fig. 5*A* and 6*A*), with a reduction in the immobilised leg ($P < 0.001$) not reflected in the control leg ($P = 0.891$). Similarly, MUP

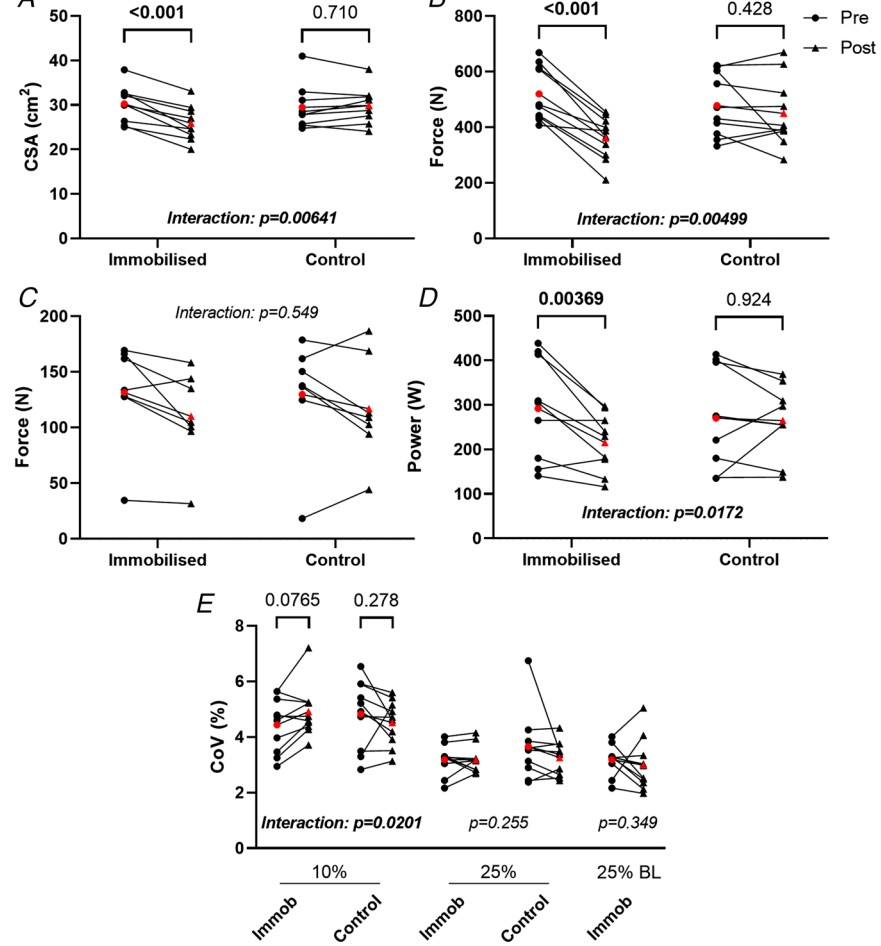

**Figure 2. Neuromuscular function before and after 15-day unilateral lower limb immobilisation in the immobilised and control legs**
*A*, vastus lateralis cross-sectional area (cm$^2$) measured by ultrasound (n = 9). *B*, knee extensor maximal voluntary isometric contraction force (N, n = 10). *C*, peak twitch force (N, n = 7) measured from maximal involuntary contraction elicited by femoral nerve stimulation. *D*, unilateral lower limb power measured using the Nottingham Power Rig (W, n = 9). *E*, knee extensor force steadiness measured during contractions at 10% and 25% relative to both follow-up and baseline maximal voluntary isometric contraction (n = 10). CoV shown as average deviation from target line (individual mean values shown pre to post in immobilised and control legs at each contraction intensity). Group mean values are shown in red. CSA; cross-sectional area, cm$^2$; centimetres squared, N; newtons, W; watts, CoV; coefficient of variation, immob; immobilised leg.

amplitude also presented a significant leg x time interaction ($P = 0.008$, Fig. 5*B* and 6*B*) with a significant reduction in the immobilised leg ($P = 0.021$) but no change in the control leg ($P = 0.634$). MUP complexity did not present a significant interaction ($P = 0.086$, Fig. 5*C* and 6*C*), nor did NMJ transmission instability ($P = 0.856$, Fig. 5*D* and 6*D*). There was a significant interaction for firing rate ($P < 0.001$, Fig. 5*E* and 6*E*), which decreased in the immobilised leg ($P < 0.001$) with no change in the control ($P = 0.482$). At 25% baseline MVC FR variability showed a significant interaction ($P < 0.001$,

Fig. 5*F* and 6*F*), which increased in the immobilised ($P < 0.001$) but not the control ($P = 0.951$) leg.

In exploratory analyses, to investigate potential relationships between neuromuscular parameters and muscle strength and size, correlation analysis was performed on the mean difference between MU characteristics assessed at 25% MVC at baseline, and 25% of FU MVC (Fig. 7). Strong correlations were observed between MUP area and amplitude ($r^2 = 0.92$, $P = 0.0003$) and between NMJ transmission instability and MVC ($r^2 = 0.89$, $P = 0.0082$). To more accurately reflect relationships

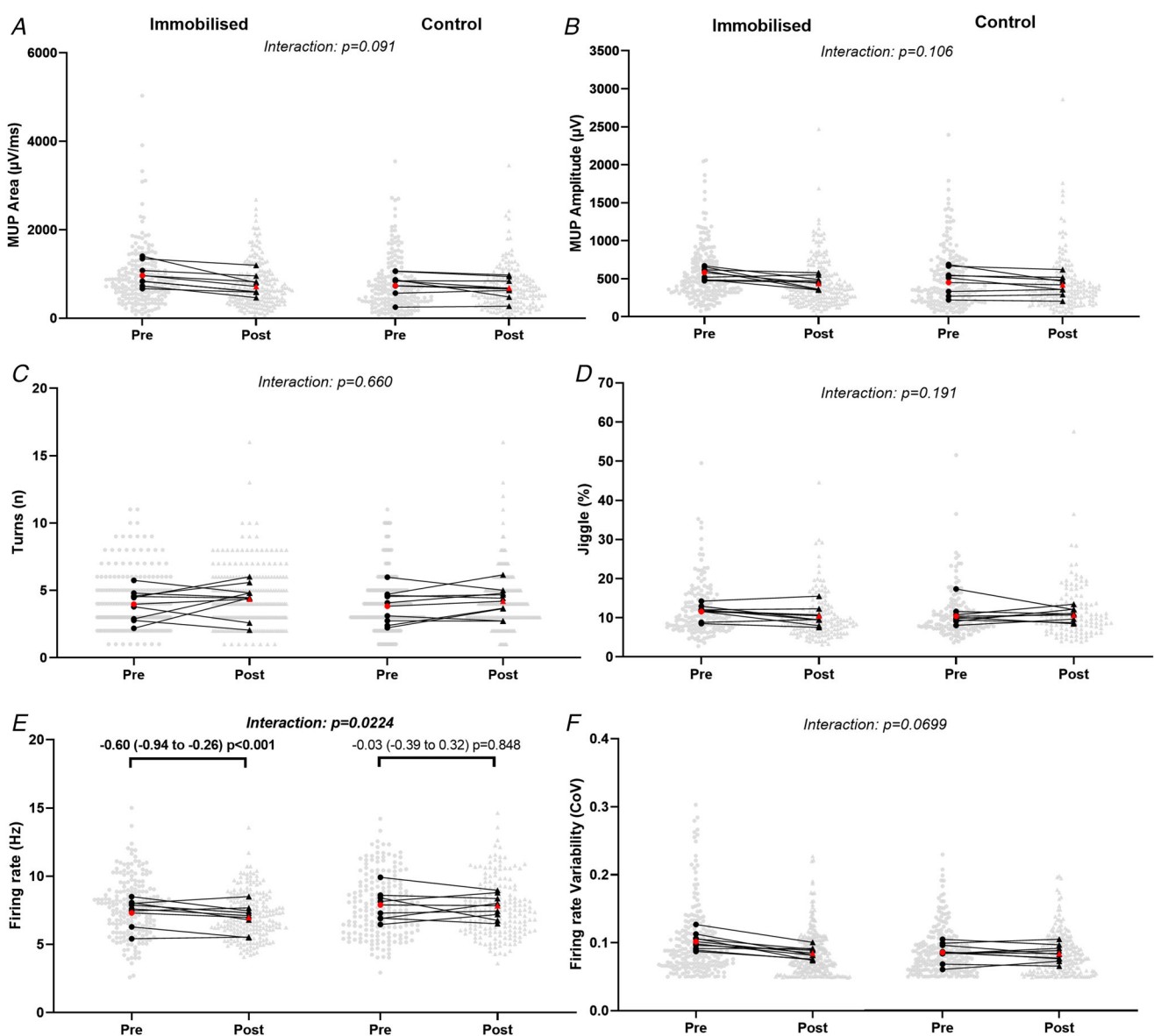

**Figure 3. Motor unit potential (MUP) and discharge characteristics of motor units sampled during contractions performed at 10% maximal voluntary force before and after 15-day unilateral leg immobilisation in both immobilised and control legs**
Data show individual mean values and pooled MUs with comparison bars showing $\beta$ coefficient and 95% CI from multi-level mixed effects models. Group mean of individual means is shown in red. N = 9. $\mu$V/ms; microvolts per millisecond, $\mu$V; microvolts, n; number, Hz; hertz, CoV; coefficient of variation.

between these variables, clustering of these variables was performed using the hierarchical clustering algorithm provided in ClustOfVar and further visualised using principal component analysis score plots (PCA, Fig. 8). The key cluster of interest consisted of MVC, CSA, NMJ transmission instability (jiggle), FR and FR variability. To test whether the neuromuscular parameters in this cluster had any influence on the changes observed in muscle strength and size, the values were first normalised to the same dynamic range before multivariate linear regression was performed. There were no significant relationships

for either MVC or CSA with NMJ transmission instability ($P = 0.191$ and $P = 0.625$ respectively), FR variability ($P = 0.759$ and $P = 0.495$) and FR ($P = 0.350$ and $P = 0.907$).

## Discussion

This study has characterised the adaptations of individual VL MUs following 15 days unilateral lower-limb immobilisation, in immobilised and non-immobilised limbs. Our findings reveal a number of adaptations

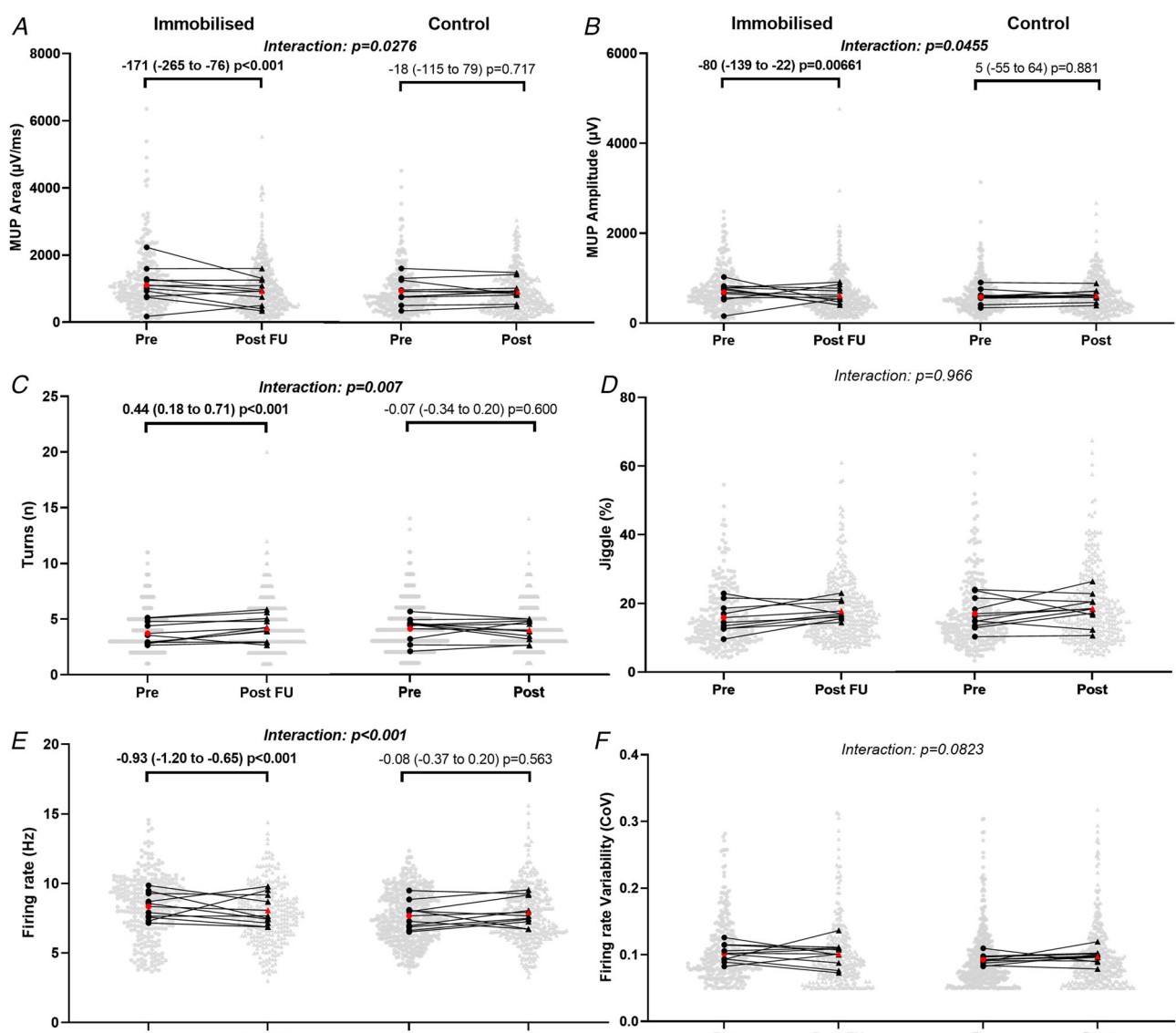

**Figure 4. Mean values of MUP and MU discharge characteristics sampled during contractions performed at 25% maximal voluntary force before and after, relative to follow-up (FU) MVC, 15-day unilateral leg immobilisation in both immobilised and control legs**
Values in the immobilised leg post-immobilisation are relative to follow-up MVC. Data show mean values and pooled MUs with comparison bars showing $\beta$ coefficient and 95% CI from multi-level mixed effects models. Group mean of individual means is shown in red. $\mu$V/ms; microvolts per millisecond, $\mu$V; microvolts, n; number, Hz; hertz, CoV; coefficient of variation.

that are unique to the immobilised limb only, with the non-immobilised limb largely unaffected. Muscle of the immobilised leg became smaller and weaker, and individual MUPs became smaller and more complex. Motor units of the immobilised leg also had reduced FR. Notably, the majority of these decrements were still apparent when force levels were normalised to that achieved prior to immobilisation, highlighting impaired neural input to muscle as a prominent contributor to the observed reduction in muscle function.

The observed decline in muscle strength (− 31%) in the immobilised limb exceeded the decline in muscle CSA

(− 15%), and this discordant finding supports several other studies employing a ULLS model. For example, 14-day unilateral knee immobilisation resulted in a ∼ 5% decrease in quadriceps CSA and a ∼ 25% decline in isometric strength (Glover et al., 2008). Additionally, following 14 days of limb cast immobilisation, an ∼ 8.5% decline in quadriceps CSA was observed alongside a ∼ 23% decline in muscle strength (Wall et al., 2014). Unilateral power output was reduced in the immobilised limb only. Bed rest studies have collectively shown a reduction in muscle power of ∼ 3% per day, although this reduction was seen to stabilise between day 5 and

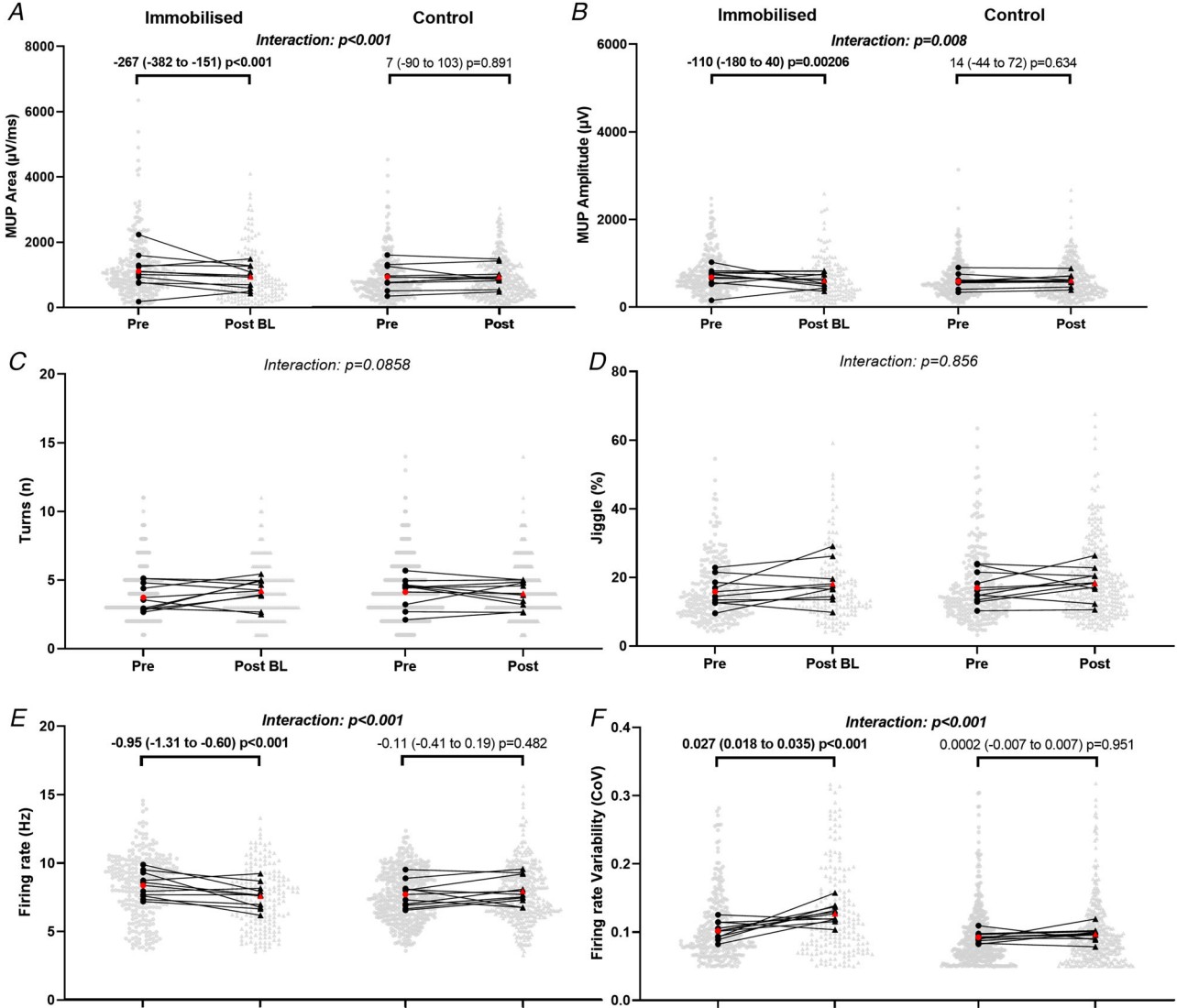

**Figure 5. Mean values of MUP parameters and MU discharge characteristics of motor units sampled during contractions performed at 25% maximal voluntary force before and after, relative to baseline (BL) MVC, 15-day unilateral leg immobilisation in both immobilised and control legs**
Values in the immobilised leg post-immobilisation are relative to baseline MVC. Data show mean values and pooled MUs with comparison bars showing $\beta$ coefficient and 95% CI from multi-level mixed effects models. Group mean of individual means is shown in red. $\mu$V/ms; microvolts per millisecond, $\mu$V; microvolts, n; number, Hz; hertz, CoV; coefficient of variation.

day 14, suggesting a sharp initial reduction (Di Girolamo et al., 2021) which may have also occurred in the present study. Furthermore, the control limb remained unaffected in terms of functional loss and clearly evidences the non-immobilised limb is spared from these decremental adaptations. Peak twitch force was unaltered when assessed with an involuntary contraction elicited from a single pulse to the femoral nerve. However, given muscle force is highly dependent on the rate of MU firing, this method may lack sensitivity when assessing involuntary force loss from a single electrical pulse. Similarly, although a leg x time interaction in force steadiness was apparent at lower contraction levels, immobilisation did not notably

influence force steadiness and indicates a preservation of basic motor control following disuse.

The size of a MUP (area and amplitude) is reflective of the depolarisation of all fibres within a single MU, within the detection area of an indwelling electrode and is proportional to the size and number of fibres contributing to it. Area and amplitude were significantly decreased post-intervention at 25% follow-up and 25% baseline MVC contraction levels in the immobilised leg only. While greater MUP size is commonly observed in aged muscle reflecting MU remodelling and reinnervation of denervated muscle fibres (Jones et al., 2022), smaller MUPs differentiated sarcopenic from non-sarcopenic

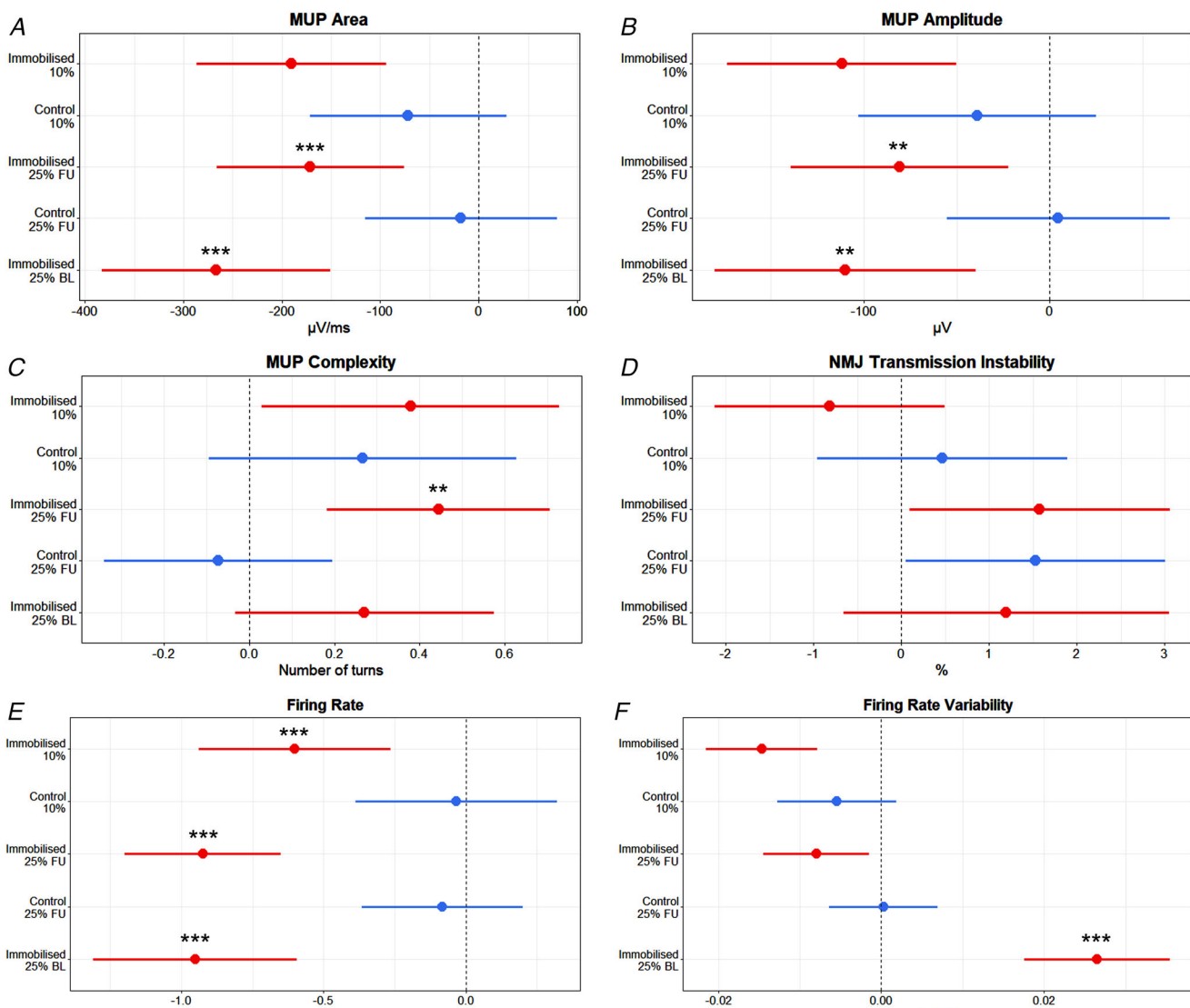

**Figure 6. Forest plots summarising model outputs from motor unit characteristics measured from the vastus lateralis during 10% MVC, and 25% normalised to follow-up (FU) and baseline (BL) MVC**
Plots display beta coefficient and 95% confidence intervals from multi-level mixed linear regression models. ** = $P < 0.01$, *** = $P < 0.001$. μV/ms; microvolts per millisecond, μV; microvolts, Hz; hertz, CoV; coefficient of variation.

older men in this muscle group (Piasecki et al., 2018). The reduction observed here may indicate partial denervation of MU fibres as a result of disuse, or, it may reflect extensive fibre atrophy across MU fibres sampled at these contraction levels. MUP size reduction has also been observed in dystrophinopathies (Zalewska et al., 2013), where amplitude in the biceps brachii was significantly below reference values in patients with Duchene and Becker muscular dystrophies (Zalewska et al., 2004). All markers of MUP size increase with increasing contraction level, reflecting the recruitment of additional, larger MUs (Guo et al., 2022), and the decreases observed in the immobilised leg were apparent at force levels normalised to maximum pre and post disuse-induced strength loss. Therefore, the reduction of force post-immobilisation cannot explain the decline in MUP size and more likely reflects a reduction in muscle fibre size and/or partial denervation/reinnervation of fibres.

Similar to MUP size, MUP complexity, defined as the number of MUP turns, was significantly greater in the immobilised leg only following the intervention, although not at all contraction levels, which is indicative of a greater temporal electrophysiological dispersion between individual fibres of the same MU (Piasecki, Garnés-Camarena et al., 2021; Stålberg & Sonoo, 1994) as a result of an increased difference in conduction times along axonal branches and/or MU fibres. Increased

MUP complexity has been observed in various myopathies which are suggestive of increased fibre diameter variability and also in neuropathies, expressing 75% greater turns than the control cohort, which is thought to be a product of the reinnervation and longer conduction times along axonal sprouts (Stewart et al., 1989). This has been reinforced specifically in myopathic conditions, as data from primarily the biceps brachii along with recordings of the vastus lateralis and gastrocnemius showed an 82% increase in polyphasic MUPs (Uncini et al., 1990). While much less severe, the changes observed here following immobilisation may suggest that some reinnervation has occurred, or alternatively that selective reduction in muscle fibre diameter has taken place, affecting MU fibres unequally resulting in more variable timing of muscle fibre action potential propagation. Although needle insertions around the muscle motor point help to minimise the effects of variable muscle fibre conduction times and enable greater focus on axonal branch conduction variability, the specific motor endplate location remains an unknown *in vivo* and these effects cannot be completely excluded.

Greater NMJ transmission instability, as quantified by NF-MUP jiggle, has been reported in patients with chronic inflammatory demyelinating polyneuropathy (Gilmore et al., 2017), diabetic neuropathy (Allen et al., 2015), and healthy ageing (Hourigan et al., 2015;

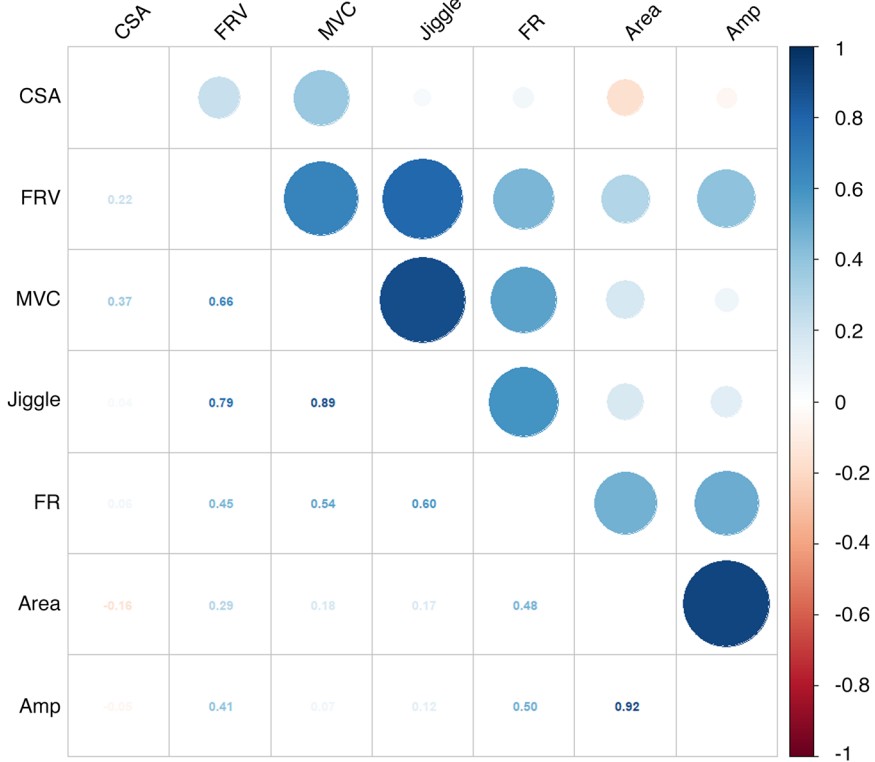

**Figure 7. Correlation matrix of relationships between eight variables measured in this study**
CSA; cross-sectional area, FRV; firing rate variability, MVC; maximal voluntary contraction, FR; firing rate, Amp; amplitude.

Piasecki, Inns et al., 2021; Piasecki, Ireland, Stashuk et al., 2016; Power et al., 2016), corresponding with larger MUPs indicative of MU remodelling (Jones et al., 2022). Here we found no statistically significant leg x time interaction in NF-MUP jiggle, at any contraction levels assessed and as such, does not support increased transmission instability at the NMJ. Recent histological findings demonstrate a greater proportion of NCAM positive fibres following 10 days bed rest which was interpreted as increased NMJ disruption (Monti et al., 2021). This method has previously been used in humans to identify denervated fibres following 3-day dry immersion (Demangel et al., 2017) and 14-day bed rest (Arentson-Lantz et al., 2016). Although the current electrophysiological findings do not support this, it may reflect differences of severity in bed rest compared to unilateral limb immobilisation. Furthermore, it is also possible disuse-induced denervation is more apparent

in later recruited MUs, beyond the range of the *in vivo* methods applied here.

MU FR was reduced following immobilisation and loss of strength and may initially be explained by the lower absolute forces produced. However, this suppression of FR was also apparent at contractions normalised to baseline strength and clearly highlights it as a contributor to functional decrements. Ionotropic synaptic inputs and neuromodulation control the excitability of motoneurons, the latter of which is largely mediated by the amplitude of persistent inward currents (PICs) which act to amplify synaptic input and are proportional to the level of localised monoamine release (Heckman et al., 2008). Although difficult to quantify *in vivo*, it is possible monoamine levels decrease in response to reduced activity, yet it is unclear how this would influence the immobilised limb only. PIC amplitudes are also sensitive to inhibition (Hyngstrom et al., 2007; Mesquita et al., 2022), and recent RNAseq data highlight increased ligand-receptor interactions between muscle and dorsal root ganglion neurons following disuse, suggestive of an increased nociceptor sensitivity and susceptibility to pain (McFarland et al., 2022). As such, it is possible increased inhibition occurred in the immobilised limb only and suppressed PIC amplitudes and MU FR, similar to that believed to explain decreased FR following knee joint trauma (Nuccio et al., 2021). No change was observed in FR variability at 10% or 25% follow-up MVC. However, when forces were normalised to MVC recorded pre-immobilisation (i.e. a greater absolute force), FR reduced to a similar extent yet with an increase of FR variability. The variability of MU FR can be reduced with training (Vila-Chã & Falla, 2016) and the current findings highlight opposing effects with muscle disuse.

Several alternative factors may also partly explain the disparity in strength and size adaptation following disuse, such as impaired calcium handling (Monti et al., 2021) suggesting a reduced efficiency of cross-bridge cycling resulting in reduced force output. Additionally, suppression of muscle protein synthesis (MPS), specifically myofibrillar proteins such as actin and myosin, is well reported as the driving mechanism of reduced muscle size during disuse (Glover et al., 2008; Nunes et al., 2022). A net negative protein balance may result in a disproportionate loss of muscle fibre contractile protein, contributing to reduced function. Although the mTOR pathway, a key driver of MPS, does not appear to be downregulated following disuse (Glover et al., 2008), others have suggested a reduction in mitochondrial protein turnover may be related to these declines in MPS during short-term immobilisation (Abadi et al., 2009) and therefore contribute to reduced muscle function. Furthermore, reduced specific tension may partly explain this disparity (Berg et al., 1997).

Additional analysis into factors potentially related to the reduction in force and changes in neuromuscular

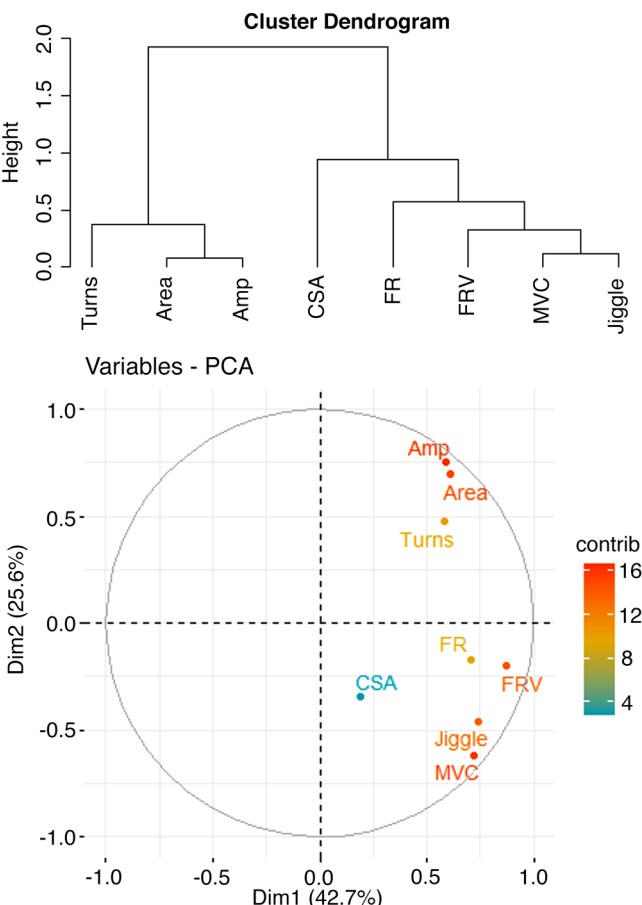

**Figure 8. Cluster tree plot (upper) and accompanying principal component analysis plot (lower) to illustrate relationships between variables within the first two dimensions of variance between variables**

Amp; amplitude, FRV; firing rate variability, MVC; maximal voluntary contraction, CSA; cross-sectional area, FR; firing rate, Dim; dimension.

parameters found a clustering of NMJ transmission instability, MU FR and FR variability with MVC and CSA. Following multivariate simple linear regression, no significant relationships were observed between either muscle size or strength with those neuromuscular parameters. This suggests that, in these data, no single variable fully explained the decline in muscle strength and size.

Since the average length of hospital stay in the United Kingdom as of 2018/19 was 4.5 days (Ewbank et al., 2020), future work in disuse should focus on the impact of such a short time frame on the neural input to muscle. However, with the increasing volume of adults >65 years old requiring short-term hospital admission (NHS Digital, 2017), this age group is also a priority for future study. Understanding these changes will provide a mechanistic basis on which to optimise rehabilitation protocols to counteract reduced muscle function.

## Limitations

The contraction levels at which motor units were sampled in the current study are of the low to mid-level and reveal nothing of adaptation to later recruited MUs. Knee extensor movements are not uniquely controlled by the VL, and although it may be a useful proxy for total quadriceps, we cannot rule out greater decrements in other muscles of this group. The current data are available in males only, and although we have highlighted similar differences across contractions in young males and females (Guo et al., 2022), there are sex-based differences in MU FR at normalised contraction levels which may respond differently to this intervention.

## Conclusion

These results support previous findings that unilateral short-term immobilisation of just 15 days leads to a decline in muscle strength unmatched by that in muscle size. Importantly, the current data highlight adaptations to neural input to muscle, evidenced by suppressed MU FR at contraction intensities relative to both the reduced maximal force following immobilisation and relative to baseline maximal force.

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

## Additional information

### Data availability statement

The data that support the findings of this study are available from the corresponding author, M. P., upon reasonable request.

### Competing interests

The authors have no competing interests to declare.

### Author contributions

T.B.I., J.J.B, E.J.O.H., P.J.A, B.E.P., and M.P. contributed to the conception and design of the work. T.B.I., J.J.B., and E.J.O.H. acquired the data. T.B.I., J.J.B., E.J.O.H., and M.P. analysed the data. T.B.I. and M.P. drafted the manuscript and prepared the figures. T.B.I., D.J.W., D.W.S., and M.P. contributed to the interpretation of the results. All authors contributed to the revision of the manuscript. All authors have approved the final version of the submitted manuscript for publication and are accountable for all aspects of the work. All persons designated as authors qualify for authorship, and all those who qualify for authorship are listed.

### Funding

This work was funded through a grant BBSRC (BB/R010358/1). This work was also supported by the Medical Research Council, United Kingdom (grant no. MR/P021220/1) as part of the MRC-Versus Arthritis Centre for Musculoskeletal Ageing Research awarded to the Universities of Nottingham and Birmingham, and the National Institute for Health Research, United Kingdom, Nottingham Biomedical Research Centre.

### Acknowledgements

The authors are grateful to all participants who took part in this research. The authors would like to thank Miss Yuxiao Guo, Miss

Isabel Ely, and Miss Eleanor Jones for their assistance with data collection.

## Keywords

electromyography, motor unit, muscle disuse, NMJ

## Supporting information

Additional supporting information can be found online in the Supporting Information section at the end of the HTML view of the article. Supporting information files available:

**Statistical Summary Document**
**Peer Review History**

