## [Peer Review History · The Journal of Physiology]

Motor unit dysregulation following 15 days of unilateral lower limb immobilisation

Thomas B. Inns, Joseph J Bass, Edward J. O. Hardy, Daniel James Wilkinson, Dan Stashuk, Philip J Atherton, Bethan E. Phillips, and Mathew Piasecki

DOI: 10.1113/JP283425

Corresponding author(s): Mathew Piasecki (mathew.piasecki@nottingham.ac.uk)

The following individual(s) involved in review of this submission have agreed to reveal their identity: Rodrigo Fernandez-Gonzalo (Referee #1); Casper Soendenbroe (Referee #2)

Review Timeline:

Submission Date:	14-Jun-2022
Editorial Decision:	01-Jul-2022
Revision Received:	01-Aug-2022
Editorial Decision:	15-Aug-2022
Revision Received:	15-Aug-2022
Accepted:	19-Aug-2022

Senior Editor: Scott Powers

Reviewing Editor: Kevin Murach

Transaction Report:

Dear Mr Inns,

Re: JP-RP-2022-283425 "Motor unit dysregulation following 15 days of unilateral lower limb immobilisation" by Thomas B. Inns, Joseph J Bass, Edward J. O. Hardy, Daniel James Wilkinson, Dan Stashuk, Philip J Atherton, Bethan E. Phillips, and Mathew Piasecki

Thank you for submitting your manuscript to The Journal of Physiology. It has been assessed by a Reviewing Editor and by 2 expert Referees and I am pleased to tell you that it is considered to be acceptable for publication following satisfactory revision.

The reports are copied at the end of this email. Please address all of the points and incorporate all requested revisions, or explain in your Response to Referees why a change has not been made.

NEW POLICY: In order to improve the transparency of its peer review process The Journal of Physiology publishes online as supporting information the peer review history of all articles accepted for publication. Readers will have access to decision letters, including all Editors' comments and referee reports, for each version of the manuscript and any author responses to peer review comments. Referees can decide whether or not they wish to be named on the peer review history document.

Authors are asked to use The Journal's premium BioRender (<https://biorender.com/>) account to create/redraw their Abstract Figures. Information on how to access The Journal's premium BioRender account is here: <https://physoc.onlinelibrary.wiley.com/journal/14697793/biorender-access> and authors are expected to use this service. This will enable Authors to download high-resolution versions of their figures. The link provided should only be used for the purposes of this submission. Authors will be charged for figures created on this premium BioRender account if they are not related to this manuscript submission.

I hope you will find the comments helpful and have no difficulty returning your revisions within 4 weeks.

Your revised manuscript should be submitted online using the links in Author Tasks Link Not Available.

Any image files uploaded with the previous version are retained on the system. Please ensure you replace or remove all files that have been revised.

REVISION CHECKLIST:

- Article file, including any tables and figure legends, must be in an editable format (eg Word)
- Abstract figure file (see above)
- Statistical Summary Document
- Upload each figure as a separate high quality file
- Upload a full Response to Referees, including a response to any Senior and Reviewing Editor Comments;
- Upload a copy of the manuscript with the changes highlighted.

- A potential 'Cover Art' file for consideration as the Issue's cover image;
- Appropriate Supporting Information (Video, audio or data set https://jp.msubmit.net/cgi-bin/main.plex?form_type=display_requirements#supp).

To create your 'Response to Referees' copy all the reports, including any comments from the Senior and Reviewing Editors, into a Word, or similar, file and respond to each point in colour or CAPITALS and upload this when you submit your revision.

I look forward to receiving your revised submission.

If you have any queries please reply to this email and staff will be happy to assist.

Yours sincerely,

Scott K. Powers
Senior Editor
The Journal of Physiology
<https://jp.msubmit.net>
<http://jp.physoc.org>
The Physiological Society
Hodgkin Huxley House
30 Farringdon Lane
London, EC1R 3AW
UK
<http://www.physoc.org>
<http://journals.physoc.org>

Please provide these items with your revision:
REQUIRED ITEMS:

-Author photo and profile. First (or joint first) authors are asked to provide a short biography (no more than 100 words for one author or 150 words in total for joint first authors) and a portrait photograph. These should be uploaded and clearly labelled with the revised version of the manuscript. See Information for Authors for further details.

-Please upload separate high-quality figure files via the submission form.

-A Statistical Summary Document, summarising the statistics presented in the manuscript, is required upon revision. It must be on the Journal's template, which can be downloaded from the link in the Statistical Summary Document section here: https://jp.msubmit.net/cgi-bin/main.plex?form_type=display_requirements#statistics

-Papers must comply with the Statistics Policy https://jp.msubmit.net/cgi-bin/main.plex?form_type=display_requirements#statistics

In summary:

-If $n \leq 30$, all data points must be plotted in the figure in a way that reveals their range and distribution. A bar graph with data points overlaid, a box and whisker plot or a violin plot (preferably with data points included) are acceptable formats.

-If $n > 30$, then the entire raw dataset must be made available either as supporting information, or hosted on a not-for-profit repository e.g. FigShare, with access details provided in the manuscript.

- n clearly defined (e.g. x cells from y slices in z animals) in the Methods. Authors should be mindful of pseudoreplication.

-All relevant n values must be clearly stated in the main text, figures and tables, and the Statistical Summary Document (required upon revision)

-The most appropriate summary statistic (e.g. mean or median and standard deviation) must be used. Standard Error of the Mean (SEM) alone is not permitted.

-Exact p values must be stated. Authors must not use 'greater than' or 'less than'. Exact p values must be stated to three significant figures even when 'no statistical significance' is claimed.

-Statistics Summary Document completed appropriately upon revision

-Please include an Abstract Figure. The Abstract Figure is a piece of artwork designed to give readers an immediate understanding of the research and should summarise the main conclusions. If possible, the image should be easily 'readable' from left to right or top to bottom. It should show the physiological relevance of the manuscript so readers can assess the importance and content of its findings. Abstract Figures should not merely recapitulate other figures in the manuscript. Please try to keep the diagram as simple as possible and without superfluous information that may distract from the main conclusion(s). Abstract Figures must be provided by authors no later than the revised manuscript stage and should

be uploaded as a separate file during online submission labelled as File Type 'Abstract Figure'. Please ensure that you include the figure legend in the main article file. All Abstract Figures should be created using BioRender. Authors should use The Journal's premium BioRender account to export high-resolution images. Details on how to use and access the premium account are included as part of this email.

-Author photo and profile. First (or joint first) authors are asked to provide a short biography (no more than 100 words for one author or 150 words in total for joint first authors) and a portrait photograph. These should be uploaded and clearly labelled with the revised version of the manuscript. See Information for Authors for further details.

-Please upload separate high-quality figure files via the submission form.

-A Statistical Summary Document, summarising the statistics presented in the manuscript, is required upon revision. It must be on the Journal's template, which can be downloaded from the link in the Statistical Summary Document section here: https://jp.msubmit.net/cgi-bin/main.plex?form_type=display_requirements#statistics

-Papers must comply with the Statistics Policy https://jp.msubmit.net/cgi-bin/main.plex?form_type=display_requirements#statistics

In summary:

-If n {less than or equal to} 30, all data points must be plotted in the figure in a way that reveals their range and distribution. A bar graph with data points overlaid, a box and whisker plot or a violin plot (preferably with data points included) are acceptable formats.

-If $n > 30$, then the entire raw dataset must be made available either as supporting information, or hosted on a not-for-profit repository e.g. FigShare, with access details provided in the manuscript.

- n clearly defined (e.g. x cells from y slices in z animals) in the Methods. Authors should be mindful of pseudoreplication.

-All relevant n values must be clearly stated in the main text, figures and tables, and the Statistical Summary Document (required upon revision)

-The most appropriate summary statistic (e.g. mean or median and standard deviation) must be used. Standard Error of the Mean (SEM) alone is not permitted.

-Exact p values must be stated. Authors must not use 'greater than' or 'less than'. Exact p values must be stated to three significant figures even when 'no statistical significance' is claimed.

-Statistics Summary Document completed appropriately upon revision

-Please include an Abstract Figure. The Abstract Figure is a piece of artwork designed to give readers an immediate understanding of the research and should summarise the main conclusions. If possible, the image should be easily 'readable' from left to right or top to bottom. It should show the physiological relevance of the manuscript so readers can assess the importance and content of its findings. Abstract Figures should not merely recapitulate other figures in the manuscript. Please try to keep the diagram as simple as possible and without superfluous information that may distract from the main conclusion(s). Abstract Figures must be provided by authors no later than the revised manuscript stage and should be uploaded as a separate file during online submission labelled as File Type 'Abstract Figure'. Please ensure that you include the figure legend in the main article file. All Abstract Figures should be created using BioRender. Authors should use The Journal's premium BioRender account to export high-resolution images. Details on how to use and access the premium account are included as part of this email.

EDITOR COMMENTS

Reviewing Editor:

Comments for Authors to ensure the paper complies with the Statistics Policy:

Precise p values required.

Comments to the Author:

Your work has been evaluated by two experts in the field. Both reviewers expressed enthusiasm for the work and agree that this manuscript could be highly influential. On balance, both reviewers also raised concerns about the statistical approach that must be addressed in order for this work to be further considered. Please address the reviewers' concerns in full and specifically focus on the issues raised regarding the statistical approach.

Senior Editor:

If the statistical summary document has errors please describe what is incorrect:

No statistical summary provided

Comments to the Author:

Thank you for submitting your work to the Journal of Physiology. Your report has been carefully evaluated by two expert referees and a review editor. While both referees find your work interesting, both reviewers has raised numerous concerns about your work that require attention during revision (see RE comments and reviewer comments for details). We look forward to receiving your revised report.

REFeree COMMENTS

Referee #1:

I congratulate the authors for an interesting study with an original set up. As the authors indicate, the impact of motor unit and/or neuromuscular junction adaptations to disuse on muscle function has been a matter of speculation for many years. Research in this area is indeed needed.

However, I have some concerns about the manuscript, particularly about the statistical approach taken by the authors, which appears to be suboptimal. As a result, it seems that the authors may have overestimated the (a priori expected) outcomes of the unloading model on MU and NMJ measurements. In the Statistical Analysis section, the authors state that 2-way analysis of variance or multi-level mixed effect linear regression models were used, and that when the leg x time interactions were not present, separate models were performed. In my view, the study design and hypothesis put forward by the authors is a classic example of two within-subject factors (leg x time) investigation, and the fact that the interaction is not significant should not lead to secondary analyses.

Furthermore, it is my understanding that post hoc tests can only be explored when significant interactions are found. The authors follow up non-significant interactions with post hoc tests for multiple variables, showing the values of the post hoc tests in the figures despite the lack of interaction. This can be somewhat misleading and could be seen as an overstatement of the results.

The final analyses including principal component analysis and multivariate linear regression lack detailed description. For example, what variables were included in each analysis, whether data from both legs were used, and what type of values (pre vs. post delta value, post results). In addition, the lack of significant interactions for most of the variables questions the use of this analysis since the variance may not come from the intervention (as both legs behave similarly, statistically speaking), but rather from other sources.

Another issue that the authors may need to briefly discuss in the manuscript is the choice of the immobilisation model over the "suspension" model. Limb immobilisation, i.e., fixing the joints at a specific angle, may affect muscle tone and cellular metabolism and even exaggerate muscle atrophy. In contrast, the free unloading of the unilateral lower limb suspension model seems to avoid such interferences and resembles "real" unloading. Although the immobilisation method may mimic a patient with a lower limb injury to a greater degree, the difference between the models should be acknowledged in the manuscript.

Following my previous comment, the authors should avoid the term unilateral lower limb suspension (ULLS), since the model they have chosen is immobilisation, not suspension.

In the results section, the smaller number of subjects analysed for some variables (i.e., VL CSA, peak twitch force, power) should be explained.

In Fig 2B, it is somewhat surprising to see that one subject experienced a massive loss of muscle force (>50%) in the control leg that was not paralleled by a similar decrease in power. I encourage the authors to carefully review these results. Could this be a measurement error?

Referee #2:

This is an interesting manuscript, from a well-established group of researchers that have previously produced several highly influential papers on human physiology in relation to ageing, exercise and disease. Certain members of the group have - it is fair to say - state-of-the-art insight into and experience with using EMG to investigate muscle activation and the interaction between skeletal muscles and the peripheral nervous system in humans. They have published using these methods several times in JoP (Piasecki et al., 2016b, 2018, 2021; Jones et al., n.d.) as well as other physiology journals (Piasecki et al., 2016a, 2019; Jones et al., 2021; Guo et al., 2021, 2022), and the overall research question in the present manuscript - disuse-induced alterations in motor unit regulation - is clearly within the scope of JoP. Importantly, another study investigating the disproportional loss of muscle force versus muscle mass was published in JoP in 2021 (Monti et al., 2021) (cited by the authors), and this present manuscript might be a valuable extension of that.

However, there are some major issues relating to statistics, reporting of data and subsequent interpretation that strongly reduces the enthusiasm at this point, and there are also several minor issues. Overall, it is this reviewer's opinion that while the study is clearly warranted within the field, substantial changes should be made throughout the manuscript. Specific feedback is provided in a section-by-section manner.

Statistics

This reviewer has serious issues with the statistics used in this manuscript.

Firstly, it should be more clearly described exactly how the multi-level mixed model was performed. Right now, we only know the name of the test, variables and the software used.

Secondly, in the statistics section, the authors write that "Where no leg x time interactions were present, separate models were performed to assess effects of time in individual limbs". However, it is not further specified which statistical test was used then? Was it simply a t-test or? Crucially, this should be COMPLETELY clear both in the statistics sections, results section and in all figures or figure legends. This is also in accordance with journal policy.

Thirdly, in continuation of the first point, it is unclear how some of the reported data can turn out to be significant? For instance, in figure 3.C a p-value of 0.022 is reported. However, when I estimate the subject-values (Immb, mean/median of 4.1/4.5 (pre) and 4.5/5 (post)) and run a simple t-test, I get a p-value of 0.197, hence nowhere near the reported one. This, most likely reflect that the values that have been used for statistical testing are not an average of measurements per subject, but rather ALL motor unit measurements pooled together, which greatly increases the n. It is not appropriate to treat a single MU as a statistical unit, instead it should be a single individual. The authors have previously published, successfully, also in JoP, using this approach (Piasecki et al., 2021; Jones et al., 2021), but that in itself is not an argument for allowing this practice. In an attempt to overcome this, I suggest that the authors, in a very clear and pedagogical manner, show the data both per subject and pooled together, perform statistical analyses of both sets of data and describe this clearly in the statistics section. This naturally means that the discussion also has to be revamped to reflect these changes.

Fourthly, the values of interactions (significant or not) should be found in the actual figure that they belong to, as this would greatly increase readability of the data/statistics.

Fifthly, please state in statistics section the type of summary data used.

Introduction

Please justify why EMG is a good model to study changes in neuromuscular innervation? This could also be done using tissue samples, as discussed in your recent review (Jones et al., n.d.).

The authors write on page 4 line 48, that disuse atrophy is "common" in clinical settings. Please specify.

Page 4 line 63-66. The authors discuss the mechanistic underpinnings of disuse atrophy, and indicate that it is less clear why function is lost (to a greater extent). However, it seems likely that an increased MPB and/or decreased MPS over time would also affect function, and therefore not be able to explain the discrepancy?

Page 5, line 70, please define "FR" on first use.

Another disuse study was published in JoP in 2021 (Monti et al., 2021), and the authors claim that structural disruption at the NMJ was observed. This is not correct. In that study CAF was used as a (tentative) blood-based biomarker (Calvani et al., 2017), NCAM as a IFF marker of denervation (Sonjak et al., 2019; Soendenbroe et al., 2021), several potential gene expression markers of denervation and activation capacity assessed using the interpolated twitch technique.

Page 5, line 76-77, the authors bring up the topic of muscle dependent disuse atrophy, yet only investigate VL in the present study. Why?

Methods

Please provide a separate section and a corresponding figure of the study design, as it, despite the simplicity of the setup, is sometimes hard to follow. Please also be consistent in the use of abbreviations such as BL and FU (see for instance fig 5), and write them out in full at first use.

There is a lack of information on the participants. Training status, smoking habits, supplement use, etc., as well as any inclusion criteria for age, height and weight. This should be specified.

Why did the authors decide to use US to measure muscle CSA, and not thigh LBM by DEXA or the gold standard VL CSA by MRI?

In general, US is very operator dependent (Sarto et al., 2021; Ritsche et al., 2021). Was it the same individual who performed all scans, and how was this standardized? This question also extends to other measurements (MVC, EMG). Did one person measure everyone or was this split between several authors?

It is said that three axial plane images were collected for US and analysed. Is this the number of measurements done per individual? If so, was a mean of the three used, the middle one, or did all three go independently into the data? Or was only one image obtained per individual?

It is not mentioned whether the investigators were blinded. Obviously, one can observe on which leg the subjects wear the cast when entering the room, but was it considered to have the operator stand outside the room when preparing the tests? Also, for some data a lot of post-processing takes place (EMG), was the investigator blinded here?

Investigating motor unit regulation under static contractions is a very interesting method, but, as the authors note themselves, they only perform low-moderate intensity contractions, meaning that in terms of fiber recruitment, it is probably only the type I fibers that are recruited. Do the authors have any data to add that could provide information on the type II fibers, perhaps rate of force/torque development (absolute or relative to MVC)? See for instance Aagaard et al., 2020 (Aagaard et al., 2002). To my knowledge, most dynamometers, even custom-built, sample more than just MVC.

Additionally, please justify why 12-15 s was chosen as duration of the submaximal contractions? Others have used longer duration (Power et al., 2012)

Were the motor point sites located identically from pre to post, and how were they located when comparing the legs at each time point?

Page 7 line 123-124, did the participants only get ONE attempt at the MVC? And if so, why? Most studies allow at least three attempts

Page 7, line 143-144, please mention the intensity of the voluntary contractions.

What software were used for DQEMG analysis?

In figure 1, please annotate within each figure what is shown. Use text and arrows. Please highlight the "turns" in F with some marker.

Also, C-E are too small to see on print. I suggest breaking A-C from D-F into two figures.

Results (figures and tables)

The number of decimals for p-values is inconsistent (see fig 3 A for example). As per journal policy there should be three significant figures: "the exact p values must be stated to three significant figures even when 'no statistical significance' is being reported".

Fig 3.C, the p value in the figure is incorrectly set as 0.02 and not 0.2 as it is in the text.

Please state the outcome of each ANOVA within each figure.

Consider going through the results of the neuromuscular parameters separate by testing conditions (10% 25%BL 25%FU) and not per test (MUP area MUPamp etc.). This would make it easier to follow as one do not have to go back and forth between figure 3 and 4.

The number of data points sometimes varies between measurements within figures. Why is that? See for instance 2.B vs 2.C. These would have been sampled at the same session in the dynamometer. For figure 3 n is said to be 9, but in 3.C only 8 dots/lines can be seen.

Fig 2-4. The colours are superfluous, simply write pre/post or group below/above.

Add mean/median line or write value.

Fig 2

Significance symbol are lacking in E (10%). According to text.

There is one major outlier in E 25% control, where the values changes from ~6.5 to ~3.5. What happened there?

Page 12 line 242, n is not specified.

Figure 4 should be separate into two figures, where FU and BL are kept separately. The connected values between FU and BL as it is right now indicating that they were not sampled at the same timepoint. Separating the data would also make it clearer that statistical testing has been done for both 25%BL and 25%FU (assumed by the reviewer). Alternatively, the same figure might be able to hold the pre values twice, so that pre-BL and pre-FU is shown separately.

Fig 5. Inconsistency in terminology. Either use MUP complexity or turns and instability or jiggle. This also goes for the rest of the manuscript.

Why is there such a large difference between 25%FU and 25%BL in F?

Fig 6. The values in yellow are not visible on print. Some of the blue ones are also very vague.

Also, the r2 value for MUP area and MUP amp is different in the text (0.846) from the figure. Is this simply a type-mistake?

Page 17 line 301-302. Please explain this better as I do not understand what is meant: "mean differences between parameters assessed at 25% pre and 25% post MVC".

Discussion

The discussion is, based on the statistical issues previously outlined, overstated and should be toned down to better reflect what can be interpreted from the data (on the level of the subject).

Multiple studies, some referenced by the authors, have provided evidence for a discordant decline in muscle mass relative to force. And neuromuscular alterations are one possible explanation. What other factors could play a role here, or do the authors believe that neuromuscular alterations can explain the entire differences?

Alterations in innervation can also be studied using other methods, and it has been in both humans (Arentson-Lantz et al., 2016; Demangel et al., 2017) and animals (Deschenes & Wilson, 2003; Baehr et al., 2016; Deschenes et al., 2021 p.202). This should at least be mentioned.

The finding of altered neuromuscular innervation following disuse has many interesting implications, but it is first and foremost of relevance in relation to hospitalization of bed-ridden patients and the substantial decrements of muscle function observed in those. This is further relevant due demographic changes that might mean that there are not enough resources to perform mobilization/exercise while hospitalized.

Despite this, the authors only briefly touch upon this in the discussion (page 22, line 444-447). On the contrary, being the devil's advocate, the data from Ewbank et al., 2020, that the authors themselves point to about length of hospital stay, are, presumably, mostly based on elderly individuals being hospitalized, and not young and healthy. As such, one could argue that the sole focus on young individuals in the present manuscript is not appropriate, given that young individuals likely regain function relatively rapidly following disuse (Suetta et al., 2013; Rejc et al., 2018). Please discuss these elements.

Page 19 line 361-362: "...and does not support the concept of a negative cross-over effect on measures of neuromuscular function". Please provide reference.

Page 19 line 363-364: "...and likely reflects the lack of sensitivity of this method". Please provide reference.

Writing / language

The manuscript could benefit from being read-through a couple of times with the perspective of improving the flow

Terminology is sometimes inconsistent. Turns and complexity as well as instability or jiggle cannot be used interchangeably.

Literature

Aagaard P, Simonsen EB, Andersen JL, Magnusson P & Dyhre-Poulsen P (2002). Increased rate of force development and neural drive of human skeletal muscle following resistance training. *Journal of Applied Physiology* 93, 1318-1326.

Arentson-Lantz EJ, English KL, Paddon-Jones D & Fry CS (2016). Fourteen days of bed rest induces a decline in satellite cell content and robust atrophy of skeletal muscle fibers in middle-aged adults. *J Appl Physiol* (1985) 120, 965-975.

Baehr LM, West DWD, Marcotte G, Marshall AG, De Sousa LG, Baar K & Bodine SC (2016). Age-related deficits in skeletal muscle recovery following disuse are associated with neuromuscular junction instability and ER stress, not impaired protein synthesis. *Aging* 8, 127-146.

de Boer MD, Maganaris CN, Seynnes OR, Rennie MJ & Narici MV (2007). Time course of muscular, neural and tendinous adaptations to 23 day unilateral lower-limb suspension in young men. *J Physiol* 583, 1079-1091.

Calvani R, Marini F, Cesari M, Tosato M, Picca A, Anker SD, von Haehling S, Miller RR, Bernabei R, Landi F, Marzetti E, & SPRINTT Consortium (2017). Biomarkers for physical frailty and sarcopenia. *Aging Clin Exp Res* 29, 29-34.

Campbell EL, Seynnes OR, Bottinelli R, McPhee JS, Atherton PJ, Jones DA, Butler-Browne G & Narici MV (2013). Skeletal muscle adaptations to physical inactivity and subsequent retraining in young men. *Biogerontology* 14, 247-259.

Demangel R, Treffel L, Py G, Brioché T, Pagano AF, Bareille M-P, Beck A, Pessemeesse L, Candau R, Gharib C, Chopard A & Millet C (2017). Early structural and functional signature of 3-day human skeletal muscle disuse using the dry immersion model. *J Physiol* 595, 4301-4315.

- Deschenes MR, Trebelhorn AM, High MC, Tufts HL & Oh J (2021). Sensitivity of subcellular components of neuromuscular junctions to decreased neuromuscular activity. *Synapse*; DOI: 10.1002/syn.22220.
- Deschenes MR & Wilson MH (2003). Age-related differences in synaptic plasticity following muscle unloading. *J Neurobiol* 57, 246-256.
- Di Girolamo FG, Fiotti N, Milanović Z, Situlin R, Mearelli F, Vinci P, Šimunič B, Pišot R, Narici M & Biolo G (2021). The Aging Muscle in Experimental Bed Rest: A Systematic Review and Meta-Analysis. *Front Nutr*; DOI: 10.3389/fnut.2021.633987.
- Guo Y, Jones EJ, Inns TB, Ely IA, Stashuk DW, Wilkinson DJ, Smith K, Piasecki J, Phillips BE, Atherton PJ & Piasecki M (2022). Neuromuscular recruitment strategies of the vastus lateralis according to sex. *Acta Physiol (Oxf)*e13803.
- Guo Y, Piasecki J, Swiecicka A, Ireland A, Phillips BE, Atherton PJ, Stashuk D, Rutter MK, McPhee JS & Piasecki M (2021). Circulating testosterone and dehydroepiandrosterone are associated with individual motor unit features in untrained and highly active older men. *Geroscience*; DOI: 10.1007/s11357-021-00482-3.
- Jones EJ, Chiou S-Y, Atherton PJ, Phillips BE & Piasecki M (n.d.). Ageing and exercise induced motor unit remodelling. *The Journal of Physiology*; DOI: 10.1113/JP281726.
- Jones EJ, Piasecki J, Ireland A, Stashuk DW, Atherton PJ, Phillips BE, McPhee JS & Piasecki M (2021). Lifelong exercise is associated with more homogeneous motor unit potential features across deep and superficial areas of vastus lateralis. *Geroscience* 43, 1555-1565.
- Jones SW, Hill RJ, Krasney PA, O'Conner B, Peirce N & Greenhaff PL (2004). Disuse atrophy and exercise rehabilitation in humans profoundly affects the expression of genes associated with the regulation of skeletal muscle mass. *FASEB J* 18, 1025-1027.
- Kawakami Y, Akima H, Kubo K, Muraoka Y, Hasegawa H, Kouzaki M, Imai M, Suzuki Y, Gunji A, Kanehisa H & Fukunaga T (2001). Changes in muscle size, architecture, and neural activation after 20 days of bed rest with and without resistance exercise. *Eur J Appl Physiol* 84, 7-12.
- Marusic U, Narici M, Simunic B, Pisot R & Ritzmann R (2021). Nonuniform loss of muscle strength and atrophy during bed rest: a systematic review. *J Appl Physiol* (1985) 131, 194-206.
- Monti E, Reggiani C, Franchi MV, Toniolo L, Sandri M, Armani A, Zampieri S, Giacomello E, Sarto F, Sirago G, Murgia M, Nogara L, Marcucci L, Ciciliot S, Šimunic B, Pišot R & Narici MV (2021). Neuromuscular junction instability and altered intracellular calcium handling as early determinants of force loss during unloading in humans. *J Physiol* 599, 3037-3061.
- Piasecki J, Inns TB, Bass JJ, Scott R, Stashuk DW, Phillips BE, Atherton PJ & Piasecki M (2021). Influence of sex on the age-related adaptations of neuromuscular function and motor unit properties in elite masters athletes. *J Physiol* 599, 193-205.
- Piasecki M, Ireland A, Coulson J, Stashuk DW, Hamilton-Wright A, Swiecicka A, Rutter MK, McPhee JS & Jones DA (2016a). Motor unit number estimates and neuromuscular transmission in the tibialis anterior of master athletes: evidence that athletic older people are not spared from age-related motor unit remodeling. *Physiological Reports* 4, e12987.
- Piasecki M, Ireland A, Piasecki J, Degens H, Stashuk DW, Swiecicka A, Rutter MK, Jones DA & McPhee JS (2019). Long-Term Endurance and Power Training May Facilitate Motor Unit Size Expansion to Compensate for Declining Motor Unit Numbers in Older Age. *Front Physiol* 10, 449.
- Piasecki M, Ireland A, Piasecki J, Stashuk DW, Swiecicka A, Rutter MK, Jones DA & McPhee JS (2018). Failure to expand the motor unit size to compensate for declining motor unit numbers distinguishes sarcopenic from non-sarcopenic older men. *J Physiol (Lond)* 596, 1627-1637.
- Piasecki M, Ireland A, Stashuk D, Hamilton-Wright A, Jones DA & McPhee JS (2016b). Age-related neuromuscular changes affecting human vastus lateralis. *The Journal of Physiology* 594, 4525-4536.
- Power GA, Dalton BH, Behm DG, Doherty TJ, Vandervoort AA & Rice CL (2012). Motor unit survival in lifelong runners is muscle dependent. *Med Sci Sports Exerc* 44, 1235-1242.
- Rejc E, Floreani M, Taboga P, Botter A, Toniolo L, Cancellara L, Narici M, Šimunič B, Pišot R, Biolo G, Passaro A, Rittweger J, Reggiani C & Lazzer S (2018). Loss of maximal explosive power of lower limbs after 2 weeks of disuse and incomplete recovery after retraining in older adults. *J Physiol* 596, 647-665.
- Ritsche P, Wirth P, Franchi MV & Faude O (2021). ACSAuto-semi-automatic assessment of human vastus lateralis and rectus femoris cross-sectional area in ultrasound images. *Sci Rep* 11, 13042.

Sarto F, Spörri J, Fitze DP, Quinlan JI, Narici MV & Franchi MV (2021). Implementing Ultrasound Imaging for the Assessment of Muscle and Tendon Properties in Elite Sports: Practical Aspects, Methodological Considerations and Future Directions. *Sports Med* 51, 1151-1170.

Soendenbroe C, Andersen JL & Mackey AL (2021). Muscle-nerve communication and the molecular assessment of human skeletal muscle denervation with aging. *American Journal of Physiology-Cell Physiology* 321, C317-C329.

Sonjak V, Jacob K, Morais JA, Rivera-Zengotita M, Spendiff S, Spake C, Taivassalo T, Chevalier S & Hepple RT (2019). Fidelity of muscle fibre reinnervation modulates ageing muscle impact in elderly women. *The Journal of Physiology* 597, 5009-5023.

Suetta C, Frandsen U, Mackey AL, Jensen L, Hvid LG, Bayer ML, Petersson SJ, Schrøder HD, Andersen JL, Aagaard P, Schjerling P & Kjaer M (2013). Ageing is associated with diminished muscle re-growth and myogenic precursor cell expansion early after immobility-induced atrophy in human skeletal muscle. *J Physiol (Lond)* 591, 3789-3804.

Suetta C, Hvid LG, Justesen L, Christensen U, Neergaard K, Simonsen L, Ortenblad N, Magnusson SP, Kjaer M & Aagaard P (2009). Effects of aging on human skeletal muscle after immobilization and retraining. *J Appl Physiol* (1985) 107, 1172-1180.

END OF COMMENTS

Confidential Review

14-Jun-2022

EDITOR COMMENTS

Reviewing Editor:

Comments for Authors to ensure the paper complies with the Statistics Policy:
Precise p values required.

Comments to the Author:

Your work has been evaluated by two experts in the field. Both reviewers expressed enthusiasm for the work and agree that this manuscript could be highly influential. On balance, both reviewers also raised concerns about the statistical approach that must be addressed in order for this work to be further considered. Please address the reviewers' concerns in full and specifically focus on the issues raised regarding the statistical approach.

Senior Editor:

If the statistical summary document has errors please describe what is incorrect:
No statistical summary provided

We have now included a statistical summary document.

Comments to the Author:

Thank you for submitting your work to the Journal of Physiology. Your report has been carefully evaluated by two expert referees and a review editor. While both referees find your work interesting, both reviewers has raised numerous concerns about your work that require attention during revision (see RE comments and reviewer comments for details). We look forward to receiving your revised report.

We are grateful to the reviewers and editors for their prompt and thorough review of our manuscript. Responses to individual comments are detailed below.

REFEREE COMMENTS

Referee #1:

I congratulate the authors for an interesting study with an original set up. As the authors indicate, the impact of motor unit and/or neuromuscular junction adaptations to disuse on

muscle function has been a matter of speculation for many years. Research in this area is indeed needed.

Many thanks. We agree this is an underexplored research area.

However, I have some concerns about the manuscript, particularly about the statistical approach taken by the authors, which appears to be suboptimal. As a result, it seems that the authors may have overestimated the (a priori expected) outcomes of the unloading model on MU and NMJ measurements. In the Statistical Analysis section, the authors state that 2-way analysis of variance or multi-level mixed effect linear regression models were used, and that when the leg x time interactions were not present, separate models were performed. In my view, the study design and hypothesis put forward by the authors is a classic example of two within-subject factors (leg x time) investigation, and the fact that the interaction is not significant should not lead to secondary analyses.

We agree the statistical approach is less than robust and have removed all analyses viewing limbs as independent measures (as per reviewer 2 comments also). All analysis is now of a 2 x 2 factorial design reflecting the comparison of adaptation of a single immobilised leg in comparison to the non-immobilised (leg x time). This outcome has altered several outcomes and we have addressed this throughout the manuscript.

Furthermore, it is my understanding that post hoc tests can only be explored when significant interactions are found. The authors follow up non-significant interactions with post hoc tests for multiple variables, showing the values of the post hoc tests in the figures despite the lack of interaction. This can be somewhat misleading and could be seen as an overstatement of the results.

As per comment above, all statistical analysis is now focussed toward the exploration of leg x time interactions. With ANOVA, main effects of time are reported where no significant interactions are present, and in multi-level mixed effects models exploring MU adaptations, interaction effects are included in all models.

The final analyses including principal component analysis and multivariate linear regression lack detailed description. For example, what variables were included in each analysis, whether data from both legs were used, and what type of values (pre vs. post delta value, post results). In addition, the lack of significant interactions for most of the variables questions the use of this analysis since the variance may not come from the intervention (as both legs behave similarly, statistically speaking), but rather from other sources.

We have added further detail to this additional exploratory analysis.

Lines 211 to 214 now include the following to clarify the variables used for the final analysis: *"The variables used for these analyses included those measured from 25% pre immobilisation and 25% FU in the immobilised limb only, namely MVC, CSA, MUP area, MUP amplitude, MUP complexity, NMJ transmission instability, MU FR and MU FR variability. Finally, using a subset of these best clustering variables, namely MVC, CSA, FR, FR variability and NMJ transmission instability multivariate linear regression was performed to determine which clustered variables best predict changes in MVC and CSA."*

Another issue that the authors may need to briefly discuss in the manuscript is the choice of the immobilisation model over the "suspension" model. Limb immobilisation, i.e., fixing the joints at a specific angle, may affect muscle tone and cellular metabolism and even exaggerate muscle atrophy. In contrast, the free unloading of the unilateral lower limb suspension model seems to avoid such interferences and resembles "real" unloading. Although the immobilisation method may mimic a patient with a lower limb injury to a greater degree, the difference between the models should be acknowledged in the manuscript.

This is a good point. We have acknowledged the difference between ULLS and limb immobilisation on lines 190 to 193 which now read:

"Although limb immobilisation has been reported to impact muscle tone and cellular metabolism in rats (Booth, 1977; Goldspink et al., 1986; Hackney & Ploutz-Snyder, 2012), it was used in favour of ULLS in this study to prevent accidental weight bearing."

Following my previous comment, the authors should avoid the term unilateral lower limb suspension (ULLS), since the model they have chosen is immobilisation, not suspension.

We agree that immobilisation is a more suitable term.

Line 190 and line 200 have been adjusted to avoid the term ULLS in favour of immobilisation.

In the results section, the smaller number of subjects analysed for some variables (i.e., VL CSA, peak twitch force, power) should be explained.

Lower n in these variables has been explained in lines 124, 130 and 157.

In Fig 2B, it is somewhat surprising to see that one subject experienced a massive loss of muscle force (>50%) in the control leg that was not paralleled by a similar decrease in power. I encourage the authors to carefully review these results. Could this be a measurement error?

We agree this is an unusual finding and have explored numerous possibilities. However, we are confident all standard operating procedures were adhered to and can find no justifiable reason to exclude the data from this participant. For interest, removing the participant from the analysis did not alter the MVC leg x time interaction effect ($p < 0.001$) nor the post hoc

comparisons of the effect of time on immobilised and control legs ($p < 0.001$ and $p = 0.956$ respectively).

Referee #2:

This is an interesting manuscript, from a well-established group of researchers that have previously produced several highly influential papers on human physiology in relation to ageing, exercise and disease. Certain members of the group have - it is fair to say - state-of-the-art insight into and experience with using EMG to investigate muscle activation and the interaction between skeletal muscles and the peripheral nervous system in humans. They have published using these methods several times in JoP (Piasecki et al., 2016b, 2018, 2021; Jones et al., n.d.) as well as other physiology journals (Piasecki et al., 2016a, 2019; Jones et al., 2021; Guo et al., 2021, 2022), and the overall research question in the present manuscript - disuse-induced alterations in motor unit regulation - is clearly within the scope of JoP. Importantly, another study investigating the disproportional loss of muscle force versus muscle mass was published in JoP in 2021 (Monti et al., 2021) (cited by the authors), and this present manuscript might be a valuable extension of that.

We are grateful for the detailed and positive summary of our work.

However, there are some major issues relating to statistics, reporting of data and subsequent interpretation that strongly reduces the enthusiasm at this point, and there are also several minor issues. Overall, it is this reviewer's opinion that while the study is clearly warranted within the field, substantial changes should be made throughout the manuscript. Specific feedback is provided in a section-by-section manner.

We agree aspects of the statistical approach are sub-optimal and have made numerous alterations. We have addressed each comment in further detail below.

Statistics

This reviewer has serious issues with the statistics used in this manuscript.

Firstly, it should be more clearly described exactly how the multi-level mixed model was performed. Right now, we only know the name of the test, variables and the software used.

We have added further detail on the multi-level mixed model inclusive of the justification of using these models, which also relates to additional comments below.

Secondly, in the statistics section, the authors write that "Where no leg x time interactions were present, separate models were performed to assess effects of time in individual limbs".

However, it is not further specified which statistical test was used then? Was it simply a t-test or? Crucially, this should be COMPLETELY clear both in the statistics sections, results section and in all figures or figure legends. This is also in accordance with journal policy.

We agree this form of analysis is less than robust and individual limbs within the same participant should not be viewed as independent factors. Primary statistical outcomes now all relate to the presence of a leg x time interaction which more suitably addresses our research questions.

Thirdly, in continuation of the first point, it is unclear how some of the reported data can turn out to be significant? For instance, in figure 3.C a p-value of 0.022 is reported. However, when I estimate the subject-values (Immb, mean/median of 4.1/4.5 (pre) and 4.5/5 (post)) and run a simple t-test, I get a p-value of 0.197, hence nowhere near the reported one. This, most likely reflect that the values that have been used for statistical testing are not an average of measurements per subject, but rather ALL motor unit measurements pooled together, which greatly increases the n. It is not appropriate to treat a single MU as a statistical unit, instead it should be a single individual. The authors have previously published, successfully, also in JoP, using this approach (Piasecki et al., 2021; Jones et al., 2021), but that in itself is not an argument for allowing this practice. In an attempt to overcome this, I suggest that the authors, in a very clear and pedagogical manner, show the data both per subject and pooled together, perform statistical analyses of both sets of data and describe this clearly in the statistics section. This naturally means that the discussion also has to be revamped to reflect these changes.

To clarify, we have not pooled MUs in these analyses or in those cited above. The n used in these analyses reflects the number of participants and *not* the total number of MUs. The pseudo-replication of data (i.e. the artificial increase of n) can be an issue with this form of data, in which multiple values are generated for each individual participant. The pooling of MUs and comparing e.g. pre to post negates the assumption of independence; put simply, 2 MUs from one individual share more in common than 2 MUs from 2 different people. A robust but somewhat crude way of negating this is to utilise mean values for each individual participant, however a more statistically appropriate way is to utilise multilevel models whereby MUs are clustered to each participant (L1; MU, L2; participant), as we have done here. This method incorporates the whole sample of extracted MUs which preserves variability within and across participants, and also provides coefficient estimates which indicate the direction and magnitude of change, with interaction effects, whilst also maintaining a true n (that of participants, not of motor units). This concept is detailed with greater clarity by Brown et al (Brown, 2021). We have expert statistical input on use of multi-level mixed effects models by Dr Andrea Venn of the University of Nottingham, and it is widely considered the most appropriate approach with this form of data, as used by ourselves and others e.g. (Mazzo *et al.*, 2021; Orsatto *et al.*, 2021).

We do agree there was a lack of clarity in our statistical description and have expanded this section, which now reads, lines 209 to 214:

"Multi-level mixed effect linear regression models were used to analyse MU parameters, in StataSE (v16.0, StataCorp LLC, TX, USA). For these models the first level was single motor unit; single motor units were clustered according to each participant to form the second level, which was defined as the participant level and reflects the total n. Two within-subject factors were included; leg (immobilised and control) and time (pre and post), and leg x time interactions were included in all models"

Fourthly, the values of interactions (significant or not) should be found in the actual figure that they belong to, as this would greatly increase readability of the data/statistics.

We agree this is more informative and have now included outputs from interaction analyses in figures 2 through 6.

Fifthly, please state in statistics section the type of summary data used.

We have added further detail on all statistical measures in the statistics section.

Introduction

Please justify why EMG is a good model to study changes in neuromuscular innervation? This could also be done using tissue samples, as discussed in your recent review (Jones et al., n.d.).

We appreciate that the use of histochemical techniques to identify muscle fibres expressing NCAM and MHCe/n etc. (Soendenbroe *et al.*, 2019) provides useful information of the innervation status of biopsied muscle, but is unable to establish if fibres belong to the same MU. We have added further justification for the use of iEMG. Lines 92 onwards

"Intramuscular EMG (iEMG) is a minimally invasive technique which enables the in-vivo recording of individual MU potentials during voluntary muscle activation, and as such has the potential to reveal functional and structural adaptations of MUs at a range of contraction levels. The purpose of this study was to quantify within-subject VL neuromuscular adaptation of immobilised and non-immobilised limbs pre and post 15-days of unilateral immobilisation."

The authors write on page 4 line 48, that disuse atrophy is "common" in clinical settings. Please specify.

We have removed the reference to clinical settings, to acknowledge this may occur in non clinical situations. We have also provided a more suitable reference. Line 55 onwards now reads:

"Disuse atrophy (DA) is the loss of skeletal muscle mass associated with decreased external loading or complete immobilisation. It commonly occurs following joint trauma, nerve injury, or prescribed bed rest (Bodine, 2013), progressing rapidly with reductions of strength occurring after just 5 days of immobilisation (Wall et al., 2014)."

Page 4 line 63-66. The authors discuss the mechanistic underpinnings of disuse atrophy, and indicate that it is less clear why function is lost (to a greater extent). However, it seems likely that an increased MPB and/or decreased MPS over time would also affect function, and therefore not be able to explain the discrepancy?

We agree this is a difficult issue to fully disentangle and even gold-standard measures of muscle size reveal little of mechanistic adaptation at the muscle fibre level (e.g. cross-bridge kinetics, calcium sensitivity, actin/myosin reductions). Members of this group have recently highlighted a minor role of MPB with muscle disuse (Brook *et al.*, 2022), and reduced MPS undoubtedly contributes to declines of size but did not match the loss of strength in only 4 days of immobilisation. Our primary aim here was to explore neural factors that may contribute to declines in strength.

Page 5, line 70, please define "FR" on first use.

Thank you. Firing rate has been defined on line 78.

Another disuse study was published in JoP in 2021 (Monti *et al.*, 2021), and the authors claim that structural disruption at the NMJ was observed. This is not correct. In that study CAF was used as a (tentative) blood-based biomarker (Calvani *et al.*, 2017), NCAM as a IFF marker of denervation (Sonjak *et al.*, 2019; Soendenbroe *et al.*, 2021), several potential gene expression markers of denervation and activation capacity assessed using the interpolated twitch technique.

This is an excellent point and we fully agree a circulating blood biomarker quantified via ELISA is not reflective of systemic NMJ function in humans. We have removed reference to CAF to more accurately reflect the indirect measures applied. Lines 79 onwards now reads:

"Findings from the VL following 10 days of bed rest showed an increased number of NCAM positive fibres, suggesting a contribution of fibre denervation following disuse (Monti et al., 2021)."

Page 5, line 76-77, the authors bring up the topic of muscle dependent disuse atrophy, yet only investigate VL in the present study. Why?

We have chosen to investigate VL due to the reported extensive atrophy when compared to other muscle groups. This has been clarified on lines 84 to 87 which now reads:

"The extent of disuse atrophy is also muscle dependent which has led to the descriptors of 'atrophy resistant' and 'atrophy susceptible' muscles (Bass et al., 2021), with the quadriceps deteriorating at a greater rate than the hamstrings following 7 days of immobilisation (Kilroe et al., 2020)"

Methods

Please provide a separate section and a corresponding figure of the study design, as it, despite the simplicity of the setup, is sometimes hard to follow. Please also be consistent in the use of abbreviations such as BL and FU (see for instance fig 5), and write them out in full at first use.

Thank you. A study design figure has been incorporated into Figure 1 to clarify this. Furthermore Figure 6 (previously 5) has been edited to contain consistent abbreviations.

There is a lack of information on the participants. Training status, smoking habits, supplement use, etc., as well as any inclusion criteria for age, height and weight. This should be specified.

We have expanded the detail on participants and line 104 onwards now reads:

"Participants aged 18 to 40 were recruited locally from the community via advertisement posters in print and on research group social media pages. Ten healthy, young male participants were recruited to take part in the study. After providing written informed consent to participate in the study, potential participants were screened for eligibility against pre-determined exclusion criteria, including body mass index (BMI) >18 or <35 kg·m², active cardiovascular, cerebrovascular, respiratory, renal, or metabolic disease, active malignancy, musculoskeletal or neurological disorders. All participants were recreationally active, and one was an active smoker, and no participants were regularly using nutritional supplements. Once eligibility was confirmed, participants were invited to the laboratory for baseline testing, as described below."

Why did the authors decide to use US to measure muscle CSA, and not thigh LBM by DEXA or the gold standard VL CSA by MRI?

Ultrasound for changes in CSA for the quadriceps are well validated against MRI, as shown in Scott *et al.*, 2017, referenced in text on line 121, during a bed rest study. Additionally, it is more feasible to use onsite and is widely accessible to other research groups.

In general, US is very operator dependent (Sarto *et al.*, 2021; Ritsche *et al.*, 2021). Was it the same individual who performed all scans, and how was this standardized? This question also extends to other measurements (MVC, EMG). Did one person measure everyone or was this split between several authors?

Line 123-124 has been added to for clarification of operators of the US:

“The same operator performed the scans at all time points for the same participant to reduce inter operator bias.”

It is said that three axial plane images were collected for US and analysed. Is this the number of measurements done per individual? If so, was a mean of the three used, the middle one, or did all three go independently into the data? Or was only one image obtained per individual?

Line 121-123 have been added to for clarification of US measurement and analysis:

“Three measurements were made from each image which were averaged, providing three mean values per participant that were subsequently averaged for each leg and timepoint for analysis.”

It is not mentioned whether the investigators were blinded. Obviously, one can observe on which leg the subjects wear the cast when entering the room, but was it considered to have the operator stand outside the room when prepare the tests? Also, for some data a lot of post-processing takes place (EMG), was the investigator blinded here?

The investigators were not blinded to ultrasound operation or analysis. EMG postprocessing was also not blinded but this was cross-checked and verified by two authors (TI and MP).

Investigating motor unit regulation under static contractions is a very interesting method, but, as the authors note themselves, they only perform low-moderate intensity contractions, meaning that in terms of fiber recruitment, it is probably only the type I fibers that are recruited. Do the authors have any data to add that could provide information on the type II fibers, perhaps rate of force/torque development (absolute or relative to MVC)? See for instance Aagaard *et al.*, 2020 (Aagaard *et al.*, 2002). To my knowledge, most dynamometers, even custom-built, sample more than just MVC.

We appreciate the reviewer raising a valid point. However, we cannot be certain that only type 1 fibres are recruited during low to moderate intensity contractions, especially as a 12

second contraction held at 25% MVC is not an easy task. We agree that a limitation of the study is that we cannot provide evidence for adaptations to later recruited MUs, as stated on lines 484-485, but it is likely that both fibre types are recruited. We are not aware of any reliable measure from a dynamometer that may predict fibre type recruitment or composition.

Additionally, please justify why 12-15 s was chosen as duration of the submaximal contractions? Others have used longer duration (Power et al., 2012).

-We commonly utilise contractions of this length to 1) minimise effects of fatigue, although we fully expect this to be minimal based on this contraction level, 2) minimise the total time of needle insertion, 3) we are confident we can sample sufficient number of MU potentials within a MU potential train to reliably identify individual MUs (e.g. a MU discharging at a rate of 10Hz would provide approx. 120 observations of the same MUP in 12s).

Were the motor point sites located identically from pre to post, and how were they located when comparing the legs at each time point?

The method of motor point identification was identical pre to post intervention. In addition, our previous findings highlight no difference in several iEMG markers when assessed at proximal and distal motor points (Piasecki et al., 2018).

Lines 201-202 now reads *"repeated in both legs with iEMG performed at the same motor point as located at the initial visit."*

Page 7 line 123-124, did the participants only get ONE attempt at the MVC? And if so, why? Most studies allow at least three attempts

Lines 138-139 have been added to clarify the number of MVC attempts:

"Three attempts were carried out and the highest value was taken as the maximal and used to determine voluntary contraction intensity."

Page 7, line 143-144, please mention the intensity of the voluntary contractions.

Lines 158-160 now reads:

"Force steadiness (FS) was assessed during the sustained voluntary contractions (described below) as the coefficient of variation of the force, averaged at each contraction intensity."

What software were used for DQEMG analysis?

DQEMG is the software program used for the analysis, which has been validated and used by ourselves and others (Stashuk, 1999; Power et al., 2016; Piasecki et al., 2021; Guo et al., 2022).

Line 173 onwards now reads:

"Decomposition-based quantitative electromyography (DQEMG) was used for all iEMG data analysis."

In figure 1, please annotate within each figure what is shown. Use text and arrows. Please highlight the "turns" in F with some marker.

Also, C-E are too small to see on print. I suggest breaking A-C from D-F into two figures.

Figure 1 has been altered to more clearly annotate what is being show in each part of the figure. Turns have been highlighted in figure 1G and the text size has been increased to improve readability.

Results (figures and tables)

The number of decimals for p-values is inconsistent (see fig 3 A for example). As per journal policy there should be three significant figures: "the exact p values must be stated to three significant figures even when 'no statistical significance' is being reported".

We thank the reviewer for pointing this out and have rectified this throughout the results section and figures.

Fig 3.C, the p value in the figure is incorrectly set as 0.02 and not 0.2 as it is in the text.

Thank you for highlighting this. This error has been corrected in fig. 3C.

Please state the outcome of each ANOVA within each figure.

All figures have been altered with Interaction effects added to each.

Consider going through the results of the neuromuscular parameters separate by testing conditions (10% \diamond 25%BL \diamond 25%FU) and not per test (MUP area \diamond MUPamp \diamond etc.). This would make it easier to follow as one do not have to go back and forth between figure 3 and 4.

We agree this will improve readability. The section titled Neuromuscular Characteristics (line 269 to 296) have been restructured to better follow the presentation of the figures, i.e., 10% for each parameter, followed by 25%FU and then 25%BL for each parameter.

The number of data points sometimes varies between measurements within figures. Why is that? See for instance 2.B vs 2.C. These would have been sampled at the same session in the

dynamometer. For figure 3 n is said to be 9, but in 3.C only 8 dots/lines can be seen.

Thank you for pointing out the missing line in fig. 3C, this was a graphical error and has been rectified. Missing n in some analyses for graphs presented in fig. 2 has been explained, as noted in an earlier comment, on lines 124, 130 and 157

Fig 2-4. The colours are superfluous, simply write pre/post or group below/above.

Figures 2-5 have been edited to remove colours. Group and time point are clearly presented below and in the top of each column respectively for figures 3-5 and are also labelled in figure 2.

Add mean/median line or write value.

Mean values for each parameter have been added as a red point in figures 2 through 5.

Fig 2

Significance symbol are lacking in E (10%). According to text.

Significance presented in fig. 2 denotes results of changes between individual legs rather than the significance of the interaction. Interaction terms have now been added to each figure.

There is one major outlier in E 25% control, where the values changes from ~6.5 to ~3.5. What happened there?

Although there is no reason to remove this outlier from the results, the removal from the model does not alter the outcome.

Page 12 line 242, n is not specified.

Lines 259 to 268 have been altered to include n for each measured parameter along with omitted units being specified

Figure 4 should be separate into two figures, where FU and BL are kept separately. The connected values between FU and BL as it is right now indicating that they were not sampled at the same timepoint. Separating the data would also make it clearer that statistical testing has been done for both 25%BL and 25%FU (assumed by the reviewer). Alternatively, the same figure might be able to hold the pre values twice, so that pre-BL and pre-FU is shown separately.

Figure 4 has been separated into figures 4 and 5 representing pre to post FU and pre to post BL respectively, reflecting the independent analyses between these groups of data. We agree this is a more useful presentation format.

Fig 5. Inconsistency in terminology. Either use MUP complexity or turns and instability or jiggle. This also goes for the rest of the manuscript.

We thank the reviewer for pointing out this inconsistency, this has been rectified throughout the manuscript.

Why is there such a large difference between 25%FU and 25%BL in F?

As noted in the Further alterations below, the coefficient of variation has now been reported rather than the FR MACD and the results section has been altered to reflect this. The new changes observed in FR variability are reported on lines 275, 285 and 293 and discussed on lines 431 to 435.

Fig 6. The values in yellow are not visible on print. Some of the blue ones are also very vague.

The matrix correlation figure represents the strength of correlation by the transparency of the data presented in the figure, and this is also replicated in opposing r^2 values within the same figure.

Also, the r^2 value for MUP area and MUP amp is different in the text (0.846) from the figure. Is this simply a type-mistake?

Thank you for pointing this out. This was a type mistake for both r^2 values in text and have been rectified on lines 328 and 329.

Page 17 line 301-302. Please explain this better as I do not understand what is meant: "mean differences between parameters assessed at 25% pre and 25% post MVC".

Lines 325 to 327 now read

"In exploratory analyses, to investigate potential relationships between neuromuscular characteristics and muscle strength and size, correlation analysis was performed on the mean difference between MU characteristics assessed at 25% MVC at baseline, and 25% of FU MVC" to add clarity to the measurements taken forward for exploratory analysis."

Discussion

The discussion is, based on the statistical issues previously outlined, overstated and should be toned down to better reflect what can be interpreted from the data (on the level of the subject).

The discussion has been altered to reflect the interpretation of data in light of the extensive statistical reanalysis.

Multiple studies, some referenced by the authors, have provided evidence for a discordant decline in muscle mass relative to force. And neuromuscular alterations are one possible explanation. What other factors could play a role here, or do the authors believe that neuromuscular alterations can explain the entire differences?

Lines 436 onwards now include a discussion of other potential factors involved in the disparity between the reduction in muscle mass and force.

“Several alternative factors may also partly explain the disparity in strength and size adaptation following disuse, such as impaired calcium handling (Monti et al., 2021) suggesting a reduced efficiency of cross-bridge cycling resulting in reduced force output. Additionally, suppression of muscle protein synthesis (MPS), specifically myofibrillar proteins such as actin and myosin, is well reported as the driving mechanism of reduced muscle size during disuse (Glover et al., 2008; Nunes et al., 2022). A net negative protein balance may result in a disproportionate loss of muscle fibre contractile protein, contributing to reduced function. Although the mTOR pathway, a key driver of MPS, does not appear to be downregulated following disuse (Glover et al., 2008), others have suggested a reduction in mitochondrial protein turnover may be related to these declines in MPS during short-term immobilisation (Abadi et al., 2009) and therefore contribute to reduced muscle function. Furthermore, reduced specific tension may partly explain this disparity (Berg et al., 1997).”

Alterations in innervation can also be studied using other methods, and it has been in both humans (Arentson-Lantz et al., 2016; Demangel et al., 2017) and animals (Deschenes & Wilson, 2003; Baehr et al., 2016; Deschenes et al., 2021 p.202). This should at least be mentioned.

Lines 409 onwards now include the following to mention alternate methods of studying innervation in humans:

“Recent histological findings demonstrate a greater proportion of NCAM positive fibres following 10 days bed rest which was interpreted as increased NMJ disruption (Monti et al., 2021). This method has previously been used in humans to identify denervated fibres following 3-day dry immersion (Demangel et al., 2017) and 14-day bed rest (Arentson-Lantz et al., 2016).”

The finding of altered neuromuscular innervation following disuse has many interesting implications, but it is first and foremost of relevance in relation to hospitalization of bed-ridden patients and the substantial decrements of muscle function observed in those. This is further relevant due demographic changes that might mean that there are not enough resources to perform mobilization/exercise while hospitalized.

Despite this, the authors only briefly touch upon this in the discussion (page 22, line 444-447). On the contrary, being the devil's advocate, the data from Ewbank et al., 2020, that the authors themselves point to about length of hospital stay, are, presumably, mostly based on elderly individuals being hospitalized, and not young and healthy. As such, one could argue that the sole focus on young individuals in the present manuscript is not appropriate, given that young individuals likely regain function relatively rapidly following disuse (Suetta et al., 2013; Rejc et al., 2018). Please discuss these elements.

Lines 454-455 now includes the following sentence to touch on the age of those requiring short-term hospital admission: "However, with the increasing volume of adults >65 years old requiring short-term hospital admission (NHS Digital, 2017), this age group is also a priority for future study."

Page 19 line 361-362: "...and does not support the concept of a negative cross-over effect on measures of neuromuscular function". Please provide reference.

To clarify the point being made here, lines 385 to 386 now reads: "in terms of functional loss and clearly evidences the non-immobilised limb is spared from these decremental adaptations."

Page 19 line 363-364: "...and likely reflects the lack of sensitivity of this method". Please provide reference.

To further clarify this point, lines 366 to 368 now reads: "However, given muscle force is highly dependent on the rate of MU firing, this method may lack sensitivity when assessing involuntary force loss from a single electrical pulse."

Writing / language

The manuscript could benefit from being read-through a couple of times with the perspective of improving the flow

Terminology is sometimes inconsistent. Turns and complexity as well as instability or jiggle cannot be used interchangeably.

Thank you for highlighting this. Inconsistency in terminology has been rectified throughout the manuscript.

Further alterations

In the initial version, FR variability was reported as the mean absolute consecutive difference (MACD) of the inter-discharge interval (IDI). This is not a widely used method therefore we have now reported the more common coefficient of variation of the IDI.

Literature

Aagaard P, Simonsen EB, Andersen JL, Magnusson P & Dyhre-Poulsen P (2002). Increased rate of force development and neural drive of human skeletal muscle following resistance training. *Journal of Applied Physiology* 93, 1318-1326.

Arentson-Lantz EJ, English KL, Paddon-Jones D & Fry CS (2016). Fourteen days of bed rest induces a decline in satellite cell content and robust atrophy of skeletal muscle fibers in middle-aged adults. *J Appl Physiol* (1985) 120, 965-975.

Baehr LM, West DWD, Marcotte G, Marshall AG, De Sousa LG, Baar K & Bodine SC (2016). Age-related deficits in skeletal muscle recovery following disuse are associated with neuromuscular junction instability and ER stress, not impaired protein synthesis. *Aging* 8, 127-146.

de Boer MD, Maganaris CN, Seynnes OR, Rennie MJ & Narici MV (2007). Time course of muscular, neural and tendinous adaptations to 23 day unilateral lower-limb suspension in young men. *J Physiol* 583, 1079-1091.

Calvani R, Marini F, Cesari M, Tosato M, Picca A, Anker SD, von Haehling S, Miller RR, Bernabei R, Landi F, Marzetti E, & SPRINTT Consortium (2017). Biomarkers for physical frailty and sarcopenia. *Aging Clin Exp Res* 29, 29-34.

Campbell EL, Seynnes OR, Bottinelli R, McPhee JS, Atherton PJ, Jones DA, Butler-Browne G & Narici MV (2013). Skeletal muscle adaptations to physical inactivity and subsequent retraining in young men. *Biogerontology* 14, 247-259.

Demangel R, Treffel L, Py G, Brioché T, Pagano AF, Bareille M-P, Beck A, Pessemeesse L, Candau R, Gharib C, Chopard A & Millet C (2017). Early structural and functional signature of 3-day human skeletal muscle disuse using the dry immersion model. *J Physiol* 595, 4301-4315.

Deschenes MR, Trebelhorn AM, High MC, Tufts HL & Oh J (2021). Sensitivity of subcellular components of neuromuscular junctions to decreased neuromuscular activity. *Synapse*; DOI: 10.1002/syn.22220.

Deschenes MR & Wilson MH (2003). Age-related differences in synaptic plasticity following muscle unloading. *J Neurobiol* 57, 246-256.

Di Girolamo FG, Fiotti N, Milanović Z, Situlin R, Mearelli F, Vinci P, Šimunič B, Pišot R, Narici M & Biolo G (2021). The Aging Muscle in Experimental Bed Rest: A Systematic Review and Meta-Analysis. *Front Nutr*; DOI: 10.3389/fnut.2021.633987.

Guo Y, Jones EJ, Inns TB, Ely IA, Stashuk DW, Wilkinson DJ, Smith K, Piasecki J, Phillips BE,

Atherton PJ & Piasecki M (2022). Neuromuscular recruitment strategies of the vastus lateralis according to sex. *Acta Physiol (Oxf)* 13803.

Guo Y, Piasecki J, Swiecicka A, Ireland A, Phillips BE, Atherton PJ, Stashuk D, Rutter MK, McPhee JS & Piasecki M (2021). Circulating testosterone and dehydroepiandrosterone are associated with individual motor unit features in untrained and highly active older men. *Geroscience*; DOI: 10.1007/s11357-021-00482-3.

Jones EJ, Chiou S-Y, Atherton PJ, Phillips BE & Piasecki M (n.d.). Ageing and exercise induced motor unit remodelling. *The Journal of Physiology*; DOI: 10.1113/JP281726.

Jones EJ, Piasecki J, Ireland A, Stashuk DW, Atherton PJ, Phillips BE, McPhee JS & Piasecki M (2021). Lifelong exercise is associated with more homogeneous motor unit potential features across deep and superficial areas of vastus lateralis. *Geroscience* 43, 1555-1565.

Jones SW, Hill RJ, Krasney PA, O'Conner B, Peirce N & Greenhaff PL (2004). Disuse atrophy and exercise rehabilitation in humans profoundly affects the expression of genes associated with the regulation of skeletal muscle mass. *FASEB J* 18, 1025-1027.

Kawakami Y, Akima H, Kubo K, Muraoka Y, Hasegawa H, Kouzaki M, Imai M, Suzuki Y, Gunji A, Kanehisa H & Fukunaga T (2001). Changes in muscle size, architecture, and neural activation after 20 days of bed rest with and without resistance exercise. *Eur J Appl Physiol* 84, 7-12.

Marusic U, Narici M, Simunic B, Pisot R & Ritzmann R (2021). Nonuniform loss of muscle strength and atrophy during bed rest: a systematic review. *J Appl Physiol* (1985) 131, 194-206.

Monti E, Reggiani C, Franchi MV, Toniolo L, Sandri M, Armani A, Zampieri S, Giacomello E, Sarto F, Sirago G, Murgia M, Nogara L, Marcucci L, Ciciliot S, Šimunic B, Pišot R & Narici MV (2021). Neuromuscular junction instability and altered intracellular calcium handling as early determinants of force loss during unloading in humans. *J Physiol* 599, 3037-3061.

Piasecki J, Inns TB, Bass JJ, Scott R, Stashuk DW, Phillips BE, Atherton PJ & Piasecki M (2021). Influence of sex on the age-related adaptations of neuromuscular function and motor unit properties in elite masters athletes. *J Physiol* 599, 193-205.

Piasecki M, Ireland A, Coulson J, Stashuk DW, Hamilton-Wright A, Swiecicka A, Rutter MK, McPhee JS & Jones DA (2016a). Motor unit number estimates and neuromuscular transmission in the tibialis anterior of master athletes: evidence that athletic older people are not spared from age-related motor unit remodeling. *Physiological Reports* 4, e12987.

Piasecki M, Ireland A, Piasecki J, Degens H, Stashuk DW, Swiecicka A, Rutter MK, Jones DA & McPhee JS (2019). Long-Term Endurance and Power Training May Facilitate Motor Unit Size

Expansion to Compensate for Declining Motor Unit Numbers in Older Age. *Front Physiol* 10, 449.

Piasecki M, Ireland A, Piasecki J, Stashuk DW, Swiecicka A, Rutter MK, Jones DA & McPhee JS (2018). Failure to expand the motor unit size to compensate for declining motor unit numbers distinguishes sarcopenic from non-sarcopenic older men. *J Physiol (Lond)* 596, 1627-1637.

Piasecki M, Ireland A, Stashuk D, Hamilton-Wright A, Jones DA & McPhee JS (2016b). Age-related neuromuscular changes affecting human vastus lateralis. *The Journal of Physiology* 594, 4525-4536.

Power GA, Dalton BH, Behm DG, Doherty TJ, Vandervoort AA & Rice CL (2012). Motor unit survival in lifelong runners is muscle dependent. *Med Sci Sports Exerc* 44, 1235-1242.

Rejc E, Floreani M, Taboga P, Botter A, Toniolo L, Cancellara L, Narici M, Šimunič B, Pišot R, Biolo G, Passaro A, Rittweger J, Reggiani C & Lazzer S (2018). Loss of maximal explosive power of lower limbs after 2 weeks of disuse and incomplete recovery after retraining in older adults. *J Physiol* 596, 647-665.

Ritsche P, Wirth P, Franchi MV & Faude O (2021). ACSAuto-semi-automatic assessment of human vastus lateralis and rectus femoris cross-sectional area in ultrasound images. *Sci Rep* 11, 13042.

Sarto F, Spörri J, Fitze DP, Quinlan JI, Narici MV & Franchi MV (2021). Implementing Ultrasound Imaging for the Assessment of Muscle and Tendon Properties in Elite Sports: Practical Aspects, Methodological Considerations and Future Directions. *Sports Med* 51, 1151-1170.

Soendenbroe C, Andersen JL & Mackey AL (2021). Muscle-nerve communication and the molecular assessment of human skeletal muscle denervation with aging. *American Journal of Physiology-Cell Physiology* 321, C317-C329.

Sonjak V, Jacob K, Morais JA, Rivera-Zengotita M, Spendiff S, Spake C, Taivassalo T, Chevalier S & Hepple RT (2019). Fidelity of muscle fibre reinnervation modulates ageing muscle impact in elderly women. *The Journal of Physiology* 597, 5009-5023.

Suetta C, Frandsen U, Mackey AL, Jensen L, Hvid LG, Bayer ML, Petersson SJ, Schrøder HD, Andersen JL, Aagaard P, Schjerling P & Kjaer M (2013). Ageing is associated with diminished muscle re-growth and myogenic precursor cell expansion early after immobility-induced atrophy in human skeletal muscle. *J Physiol (Lond)* 591, 3789-3804.

Suetta C, Hvid LG, Justesen L, Christensen U, Neergaard K, Simonsen L, Ortenblad N, Magnusson SP, Kjaer M & Aagaard P (2009). Effects of aging on human skeletal muscle after

immobilization and retraining. *J Appl Physiol* (1985) 107, 1172-1180.

References

- Arentson-Lantz EJ, English KL, Paddon-Jones D & Fry CS (2016). Fourteen days of bed rest induces a decline in satellite cell content and robust atrophy of skeletal muscle fibers in middle-aged adults. *J Appl Physiol* **120**, 965–975. DOI: 10.1152/jappphysiol.00799.2015.
- Bass JJ, Hardy EJO, Inns TB, Wilkinson DJ, Piasecki M, Morris RH, Spicer A, Sale C, Smith K, Atherton PJ & Phillips BE (2021). Atrophy Resistant vs. Atrophy Susceptible Skeletal Muscles: “aRaS” as a Novel Experimental Paradigm to Study the Mechanisms of Human Disuse Atrophy. *Front Physiol* **12**, 1–11. DOI: 10.3389/fphys.2021.653060.
- Bodine SC (2013). Disuse-induced muscle wasting. *Int J Biochem Cell Biol* **45**, 2200–2208. DOI: 10.1016/j.biocel.2013.06.011.
- Brook MS, Stokes T, Gorissen SHM, Bass JJ, McGlory C, Cegielski J, Wilkinson DJ, Phillips BE, Smith K, Phillips SM & Atherton PJ (2022). Declines in muscle protein synthesis account for short-term muscle disuse atrophy in humans in the absence of increased muscle protein breakdown. *J Cachexia Sarcopenia Muscle* DOI: 10.1002/jcsm.13005.
- Brown VA (2021). An Introduction to Linear Mixed-Effects Modeling in R. *Adv Methods Pract Psychol Sci* DOI: 10.1177/2515245920960351.
- Demangel R, Treffel L, Py G, Brioché T, Pagano AF, Bareille MP, Beck A, Pessemeesse L, Candau R, Gharib C, Chopard A & Millet C (2017). Early structural and functional signature of 3-day human skeletal muscle disuse using the dry immersion model. *J Physiol* **595**, 4301–4315. DOI: 10.1113/JP273895.
- Guo Y, Jones EJ, Inns TB, Ely IA, Stashuk DW, Wilkinson DJ, Smith K, Piasecki J, Phillips BE, Atherton PJ & Piasecki M (2022). Neuromuscular recruitment strategies of the vastus lateralis according to sex. *Acta Physiol* 1–14. DOI: 10.1111/apha.13803.
- Kilroe SP, Fulford J, Jackman SR, Van Loon LJC & Wall BT (2020). Temporal Muscle-specific Disuse Atrophy during One Week of Leg Immobilization. *Med Sci Sports Exerc* **52**, 944–954. DOI: 10.1249/MSS.0000000000002200.
- Mazzo MR, Weinman LE, Giustino V, Mclagan B, Maldonado J & Enoka RM (2021). Changes in neural drive to calf muscles during steady submaximal contractions after repeated static stretches. *J Physiol* **599**, 4321–4336. DOI: 10.1113/JP281875.
- Monti E, Reggiani C, Franchi M V., Toniolo L, Sandri M, Armani A, Zampieri S, Giacomello E, Sarto F, Sirago G, Murgia M, Nogara L, Marcucci L, Ciciliot S, Šimunic B, Pišot R & Narici M V. (2021). Neuromuscular junction instability and altered intracellular calcium handling as early determinants of force loss during unloading in humans. *J Physiol* **599**, 3037–3061. DOI: 10.1113/JP281365.
- NHS Digital (2017). Hospital Admitted Patient Care Activity 2016-17. *Hosp Admit Patient Care Act* 1–35. .
- Orssatto LBR, Borg DN, Blazeovich AJ, Sakugawa RL, Shield AJ & Trajano GS (2021). Intrinsic motoneuron excitability is reduced in soleus and tibialis anterior of older adults. *GeroScience* **43**, 2719–2735. DOI: 10.1007/s11357-021-00478-z.

- Piasecki J, Inns TB, Bass JJ, Scott R, Stashuk DW, Phillips BE, Atherton PJ & Piasecki M (2021). Influence of sex on the age-related adaptations of neuromuscular function and motor unit properties in elite masters athletes. *J Physiol* **599**, 193–205. DOI: 10.1113/JP280679.
- Piasecki M, Ireland A, Piasecki J, Stashuk DW, McPhee JS & Jones DA (2018). The reliability of methods to estimate the number and size of human motor units and their use with large limb muscles. *Eur J Appl Physiol* **118**, 767–775. DOI: 10.1007/s00421-018-3811-5.
- Power GA, Allen XMD, Gilmore KJ, Stashuk DW, Doherty TJ, Hepple RT, Taivassalo T & Rice CL (2016). Motor unit number and transmission stability in octogenarian world class athletes: Can age-related deficits be outrun? *J Appl Physiol* **121**, 1013–1020. DOI: 10.1152/jappphysiol.00149.2016.
- Scott JM, Martin DS, Ploutz-Snyder R, Matz T, Caine T, Downs M, Hackney K, Buxton R, Ryder JW & Ploutz-Snyder L (2017). Panoramic ultrasound: a novel and valid tool for monitoring change in muscle mass. *J Cachexia Sarcopenia Muscle* **8**, 475–481. DOI: 10.1002/jcsm.12172.
- Soendenbroe C, Heisterberg MF, Schjerling P, Karlsen A, Kjaer M, Andersen JL & Mackey AL (2019). Molecular indicators of denervation in aging human skeletal muscle. *Muscle and Nerve* **60**, 453–463. DOI: 10.1002/mus.26638.
- Stashuk DW (1999). Detecting single fiber contributions to motor unit action potentials. *Muscle and Nerve* **22**, 218–229. DOI: 10.1002/(SICI)1097-4598(199902)22:2<218::AID-MUS10>3.0.CO;2-S.
- Wall BT, Dirks ML, Snijders T, Senden JMG, Dolmans J & Van Loon LJC (2014). Substantial skeletal muscle loss occurs during only 5 days of disuse. *Acta Physiol* **210**, 600–611. DOI: 10.1111/apha.12190.

Dear Mr Inns,

Re: JP-RP-2022-283425R1 "Motor unit dysregulation following 15 days of unilateral lower limb immobilisation" by Thomas B. Inns, Joseph J Bass, Edward J. O. Hardy, Daniel James Wilkinson, Dan Stashuk, Philip J Atherton, Bethan E. Phillips, and Mathew Piasecki

Thank you for submitting your revised Research Article to The Journal of Physiology. It has been assessed by the original Reviewing Editor and Referees and has been well received. Some final revisions have been requested.

The reports are copied at the end of this email. Please address all of the points and incorporate all requested revisions, or explain in your Response to Referees why a change has not been made.

NEW POLICY: In order to improve the transparency of its peer review process The Journal of Physiology publishes online as supporting information the peer review history of all articles accepted for publication. Readers will have access to decision letters, including all Editors' comments and referee reports, for each version of the manuscript and any author responses to peer review comments. Referees can decide whether or not they wish to be named on the peer review history document.

Authors are asked to use The Journal's premium BioRender (<https://biorender.com/>) account to create/redraw their Abstract Figures. Information on how to access The Journal's premium BioRender account is here: <https://physoc.onlinelibrary.wiley.com/journal/14697793/biorender-access> and authors are expected to use this service. This will enable Authors to download high-resolution versions of their figures. The link provided should only be used for the purposes of this submission. Authors will be charged for figures created on this premium BioRender account if they are not related to this manuscript submission.

I hope you will find the comments helpful and have no difficulty returning your revisions within 4 weeks.

Your revised manuscript should be submitted online using the links in Author Tasks Link Not Available.

Any image files uploaded with the previous version are retained on the system. Please ensure you replace or remove all files that have been revised.

REVISION CHECKLIST:

- Article file, including any tables and figure legends, must be in an editable format (eg Word)
- Abstract figure file (see above)
- Statistical Summary Document
- Upload each figure as a separate high quality file
- Upload a full Response to Referees, including a response to any Senior and Reviewing Editor Comments;
- Upload a copy of the manuscript with the changes highlighted.

- A potential 'Cover Art' file for consideration as the Issue's cover image;
- Appropriate Supporting Information (Video, audio or data set https://jp.msubmit.net/cgi-bin/main.plex?form_type=display_requirements#supp).

To create your 'Response to Referees' copy all the reports, including any comments from the Senior and Reviewing Editors, into a Word, or similar, file and respond to each point in colour or CAPITALS and upload this when you submit your revision.

I look forward to receiving your revised submission.

If you have any queries please reply to this email and staff will be happy to assist.

Yours sincerely,

Scott K. Powers
Senior Editor
The Journal of Physiology
<https://jp.msubmit.net>
<http://jp.physoc.org>
The Physiological Society
Hodgkin Huxley House
30 Farringdon Lane
London, EC1R 3AW
UK
<http://www.physoc.org>
<http://journals.physoc.org>

REQUIRED ITEMS:

-Papers must comply with the Statistics Policy https://jp.msubmit.net/cgi-bin/main.plex?form_type=display_requirements#statistics

In summary:

-If n {less than or equal to} 30, all data points must be plotted in the figure in a way that reveals their range and distribution. A bar graph with data points overlaid, a box and whisker plot or a violin plot (preferably with data points included) are acceptable formats.

-If $n > 30$, then the entire raw dataset must be made available either as supporting information, or hosted on a not-for-profit repository e.g. FigShare, with access details provided in the manuscript.

-' n ' clearly defined (e.g. x cells from y slices in z animals) in the Methods. Authors should be mindful of pseudoreplication.

-All relevant ' n ' values must be clearly stated in the main text, figures and tables, and the Statistical Summary Document (required upon revision)

-The most appropriate summary statistic (e.g. mean or median and standard deviation) must be used. Standard Error of the Mean (SEM) alone is not permitted.

-Exact p values must be stated. Authors must not use 'greater than' or 'less than'. Exact p values must be stated to three significant figures even when 'no statistical significance' is claimed.

EDITOR COMMENTS

Reviewing Editor:

Both reviewers noted a substantial improvement over the original submissions. I a few minor comments remain to be addressed, but once these are managed, we can make a timely decision.

Senior Editor:

Thank you for revising your manuscript according to reviewer suggestions. Both reviewers are pleased with your revisions; however, reviewer 1 has provided a few additional minor suggestions to further improve the manuscript prior to acceptance. Therefore, please consider reviewer 1 suggestions when revising your work. We look forward to receiving your revised manuscript.

REFEREE COMMENTS

Referee #1:

First, I would like to congratulate the authors on a much improved version of the manuscript. In my opinion, the main limitation highlighted by myself and the other reviewer (i.e., the statistical approach) has been addressed. The article is now more balanced in terms of actual results and discussion/conclusion. Good job.

I have some minor comments and suggestions that the authors could consider to improve the reading flow and make the manuscript easier to understand.

Abstract

Line 42 & 45: Spell out EMG and MUs since you are using them for the first time.

Line 48: Avoid using VL in this context because loss of muscle strength does not occur only in that specific muscle. Use limb or knee extensor muscles.

In describing the neuromuscular results, it would be helpful to add some numbers (perhaps %) to give an idea of the magnitude of the changes.

Line 52: I assume discharge rate is used as a synonym for firing rate. I would suggest being consistent throughout the manuscript (i.e., always use firing rate).

Introduction

Line 55: Delete DA, as it does not appear to be used in any other section of the manuscript.

Lines 84-87: While the authors justify the use of the quadriceps muscles over the hamstrings, I think the real justification should be in the choice of the quadriceps muscle group over the calf muscles. The calf muscles are more susceptible to the consequences of unloading, as has been shown in several unloading models (e.g., Alkner & Tesch EJAP 93(3):294-305;2004), and thus may have been a better "atrophy susceptible" muscle target for the current experiments.

Methods

Line 115: the abbreviation VL has already been introduced

Lines 124-125: insert "to" between scans and reduce

Lines 125, 132, 158-159: right now these sentences starting with N=x do not contribute to a good reading flow.

Line 134: add information about the dynamometer used (at least the main components, since it is a custom-built device)

Line 188: FR has already been introduced.

Line 216: be consistent in using N or n for the number of subjects

Line 227: I would suggest avoiding the use of FU as an abbreviation for follow-up in the text - see, e.g., lines 330, 378 (though it may be useful in figures). The manuscript already contains numerous abbreviations and adding additional (not commonly used) abbreviations complicates flow and understanding. The same is true for BL, which is used in line 261 (and 378) but is not written out beforehand.

Figure 1D: The arrow points to the vastus intermedius muscle, not the lateralis muscle. Correct (it is problematic when seen in black and white).

Results

Line 255 & Figure 2C: The p-value is not the same. Correct.

Line 298: Firing rate - > FR

Discussion

Lines 359-364: This is too long a sentence. I would suggest splitting it into (at least) 2 sentences and starting the second with "For example, 14 days of....". Right now, it does not read well.

Lines 364-367: Is the information presented here relevant to the current study? I would argue that the fact that it is a different unloading model, it does not directly affect the results of the current study, and it is something that is not being studied here makes it unnecessary.

Line 414: all -> any

Conclusion

The last part of the conclusion "identifies interventional targets for cases of muscle disuse" is unclear. Does the current study design allow for the identification of "interventional targets"? I would argue that it does not. However, if the authors believe that it does, I encourage them to explain how, and to elaborate on the meaning of this statement.

Referee #2:

The authors have done very well to answer all my questions and comments with appropriate changes. And as a result, I believe, that the manuscript is now in a better and more solid state. I remain sceptical about the statistical approach for the MU measurements, but the authors go to great extent to explain and justify their decision, which is good.

END OF COMMENTS

1st Confidential Review

01-Aug-2022

Re: JP-RP-2022-283425R1

EDITOR COMMENTS

Reviewing Editor:

Both reviewers noted a substantial improvement over the original submissions. A few minor comments remain to be addressed, but once these are managed, we can make a timely decision.

Senior Editor:

Thank you for revising your manuscript according to reviewer suggestions. Both reviewers are pleased with your revisions; however, reviewer 1 has provided a few additional minor suggestions to further improve the manuscript prior to acceptance. Therefore, please consider reviewer 1 suggestions when revising your work. We look forward to receiving your revised manuscript.

We again thank the reviewers and editors for the swift response, and we agree the additional comments further improve the manuscript.

REFEREE COMMENTS

Referee #1:

First, I would like to congratulate the authors on a much improved version of the manuscript. In my opinion, the main limitation highlighted by myself and the other reviewer (i.e., the statistical approach) has been addressed. The article is now more balanced in terms of actual results and discussion/conclusion. Good job.

I have some minor comments and suggestions that the authors could consider to improve the reading flow and make the manuscript easier to understand.

Abstract

Line 42 & 45: Spell out EMG and MUs since you are using them for the first time.

On lines 43 and 46 these abbreviations have now been spelled out.

Line 48: Avoid using VL in this context because loss of muscle strength does not occur only in that specific muscle. Use limb or knee extensor muscles.

We thank the reviewer for highlighting this, and on line 49 VL has been changed to limb.

In describing the neuromuscular results, it would be helpful to add some numbers (perhaps %) to give an idea of the magnitude of the changes.

In the abstract we have now included % change based on beta values for parameters of MU size and MU firing rate. Line 49 onwards now reads:

Parameters of MUP size were reduced by 11 to 24% with immobilisation, while neuromuscular junction (NMJ) transmission instability remained unchanged, and MU firing rate decreased by 8 to 11% at several contraction levels.

Line 52: I assume discharge rate is used as a synonym for firing rate. I would suggest being consistent throughout the manuscript (i.e., always use firing rate).

We agree consistency is preferential and have altered this. Line 52 onwards now reads

"These findings highlight impaired neural input following immobilisation reflected by suppressed MU firing rate which may underpin the disproportionate reductions of strength relative to muscle size."

Introduction

Line 55: Delete DA, as it does not appear to be used in any other section of the manuscript.

DA has been removed from line 56.

Lines 84-87: While the authors justify the use of the quadriceps muscles over the hamstrings, I think the real justification should be in the choice of the quadriceps muscle group over the calf muscles. The calf muscles are more susceptible to the consequences of unloading, as has been shown in several unloading models (e.g., Alkner & Tesch EJAP 93(3):294-305;2004), and thus may have been a better "atrophy susceptible" muscle target for the current experiments.

We fully agree that the calf muscles are the most appropriate atrophy susceptible muscle

target for the current experiments. However the purpose of this study was to study a larger, widely studied and functionally important muscle.

Methods

Line 115: the abbreviation VL has already been introduced

Vastus lateralis has been removed from line 115.

Lines 124-125: insert "to" between scans and reduce

This change has been made on line 124.

Lines 125, 132, 158-159: right now these sentences starting with N=x do not contribute to a good reading flow.

We thank the reviewer for raising this. Lines 124-125, 131-132 and 159-160 have been altered to read "*Due to equipment malfunction at follow-up, n=9 for US.*",

"Due to equipment malfunction at follow-up, n=9 for unilateral lower limb power assessment.", and

"As three participants were unable to tolerate femoral nerve stimulation at follow-up visits, n=7." respectively.

Line 134: add information about the dynamometer used (at least the main components, since it is a custom-built device)

We have added further information on the dynamometer. Line 134 onwards now reads:

"Participants were seated in a custom-built isometric dynamometer (Load cell amplifier; LCA1, 12V 1A medical PSU, GDM25B12-P1J, Sunpower Electronics, Reading, UK) with their knee joint angle fixed at 90° while the hip joint angle was at 110°."

Line 188: FR has already been introduced.

Line 188 onwards now reads:

"MU FR was recorded as the rate of occurrence per second of MUPs within a MUPT in Hz, and MU FR variability was determined as the coefficient of variation of the inter-discharge interval."

Line 216: be consistent in using N or n for the number of subjects

We appreciate the reviewer highlighting this. N was capitalised in lines 125, 132, 159-160 due to it being at the beginning of a sentence rather than the use of n on line 215 being mid-way through. However as addressed in a previous comment, lines 125, 132, 159-160

have been altered and now use n.

Line 227: I would suggest avoiding the use of FU as an abbreviation for follow-up in the text - see, e.g., lines 330, 378 (though it may be useful in figures). The manuscript already contains numerous abbreviations and adding additional (not commonly used) abbreviations complicates flow and understanding. The same is true for BL, which is used in line 261 (and 378) but is not written out beforehand.

We agree that this increases complication and FU abbreviations have been replaced on lines 226 and 376 while BL has been replaced on lines 260 and 376.

Figure 1D: The arrow points to the vastus intermedius muscle, not the lateralis muscle. Correct (it is problematic when seen in black and white).

We thank the reviewer for highlighting this and have extended the arrow to point to the vastus lateralis more clearly.

Results

Line 255 & Figure 2C: The p-value is not the same. Correct.

This type error has been corrected on line 254.

Line 298: Firing rate - > FR

This change has been made on line 295.

Discussion

Lines 359-364: This is too long a sentence. I would suggest splitting it into (at least) 2 sentences and starting the second with "For example, 14 days of....". Right now, it does not read well.

Line 359 onwards now reads:

"For example, 14-day unilateral knee immobilisation resulted in a ~5% decrease in quadriceps CSA and a ~25% decline in isometric strength (Glover et al., 2008). Additionally, following 14 days of limb cast immobilisation, an ~8.5% decline in quadriceps CSA was observed alongside a ~23% decline in muscle strength (Wall et al., 2014)."

"

Lines 364-367: Is the information presented here relevant to the current study? I would argue that the fact that it is a different unloading model, it does not directly affect the results of the current study, and it is something that is not being studied here makes it unnecessary.

We appreciate the reviewer's suggestion. However we believe this information is relevant to the current study as it discusses the reduction in power observed over time during a situation of disuse atrophy. We appreciate that, as discussed later in the paper, that bed rest is a more severe model of disuse atrophy. Line 362 now reads "*Bed rest studies have collectively shown a reduction in muscle power...*" to clarify the relevance to the current study's assessment of muscle power.

Line 414: all -> any

Corrected.

Conclusion

The last part of the conclusion "identifies interventional targets for cases of muscle disuse" is unclear. Does the current study design allow for the identification of "interventional targets"? I would argue that it does not. However, if the authors believe that it does, I encourage them to explain how, and to elaborate on the meaning of this statement.

We agree discussing the plasticity of MU firing rate is beyond the reach of this conclusion and further data would better support it as a possible interventional target. We have removed this sentence from the manuscript.

Referee #2:

The authors have done very well to answer all my questions and comments with appropriate changes. And as a result, I believe, that the manuscript is now in a better and more solid state. I remain sceptical about the statistical approach for the MU measurements, but the authors go to great extent to explain and justify their decision, which is good.

We would like to thank the reviewer for their comment and appreciate their input towards improving the manuscript. We are confident the statistical approach used here is robust.

END OF COMMENTS

Dear Dr Piasecki,

Re: JP-RP-2022-283425R2 "Motor unit dysregulation following 15 days of unilateral lower limb immobilisation" by Thomas B. Inns, Joseph J Bass, Edward J. O. Hardy, Daniel James Wilkinson, Dan Stashuk, Philip J Atherton, Bethan E. Phillips, and Mathew Piasecki

I am pleased to tell you that your paper has been accepted for publication in The Journal of Physiology.

NEW POLICY: In order to improve the transparency of its peer review process The Journal of Physiology publishes online as supporting information the peer review history of all articles accepted for publication. Readers will have access to decision letters, including all Editors' comments and referee reports, for each version of the manuscript and any author responses to peer review comments. Referees can decide whether or not they wish to be named on the peer review history document.

The last Word version of the paper submitted will be used by the Production Editors to prepare your proof. When this is ready you will receive an email containing a link to Wiley's Online Proofing System. The proof should be checked and corrected as quickly as possible.

Authors should note that it is too late at this point to offer corrections prior to proofing. The accepted version will be published online, ahead of the copy edited and typeset version being made available. Major corrections at proof stage, such as changes to figures, will be referred to the Reviewing Editor for approval before they can be incorporated. Only minor changes, such as to style and consistency, should be made a proof stage. Changes that need to be made after proof stage will usually require a formal correction notice.

All queries at proof stage should be sent to TJP@wiley.com

Are you on Twitter? Once your paper is online, why not share your achievement with your followers. Please tag The Journal (@jphysiol) in any tweets and we will share your accepted paper with our 23,000+ followers!

Yours sincerely,

Scott K. Powers
Senior Editor
The Journal of Physiology
<https://jp.msubmit.net>
<http://jp.physoc.org>
The Physiological Society
Hodgkin Huxley House
30 Farringdon Lane
London, EC1R 3AW
UK
<http://www.physoc.org>
<http://journals.physoc.org>

P.S. - You can help your research get the attention it deserves! Check out Wiley's free Promotion Guide for best-practice recommendations for promoting your work at www.wileyauthors.com/eeo/guide. And learn more about Wiley Editing Services which offers professional video, design, and writing services to create shareable video abstracts, infographics, conference posters, lay summaries, and research news stories for your research at www.wileyauthors.com/eeo/promotion.

*** IMPORTANT NOTICE ABOUT OPEN ACCESS ***

To assist authors whose funding agencies mandate public access to published research findings sooner than 12 months after publication The Journal of Physiology allows authors to pay an open access (OA) fee to have their papers made freely available immediately on publication.

You will receive an email from Wiley with details on how to register or log-in to Wiley Authors Services where you will be able to place an OnlineOpen order.

You can check if your funder or institution has a Wiley Open Access Account here <https://authorservices.wiley.com/author-resources/Journal-Authors/licensing-and-open-access/open-access/author-compliance-tool.html>

Your article will be made Open Access upon publication, or as soon as payment is received.

If you wish to put your paper on an OA website such as PMC or UKPMC or your institutional repository within 12 months of publication you must pay the open access fee, which covers the cost of publication.

OnlineOpen articles are deposited in PubMed Central (PMC) and PMC mirror sites. Authors of OnlineOpen articles are permitted to post the final, published PDF of their article on a website, institutional repository, or other free public server, immediately on publication.

Note to NIH-funded authors: The Journal of Physiology is published on PMC 12 months after publication, NIH-funded authors DO NOT NEED to pay to publish and DO NOT NEED to post their accepted papers on PMC.

EDITOR COMMENTS

Reviewing Editor:

Both reviewers are satisfied with your responses. Thank you for submitting your best work to The Journal of Physiology.

Senior Editor:

Thank you for submitting this outstanding report to the Journal of Physiology and congratulations on the acceptance of your manuscript for publication in the journal!

REFEREE COMMENTS

Referee #1:

The authors have addressed all the issues. I have no further comments.

Referee #2:

I do not have anything to add at this point. The manuscript overall is substantially improved from the first draft.